# Unveiling chemotherapy-induced immune landscape remodeling and metabolic reprogramming in lung adenocarcinoma by scRNA-sequencing

Yiwei Huang[1†], Gujie Wu[1†], Guoshu Bi[1†], Lin Cheng[2†], Jiaqi Liang[1], Ming Li[1], Huan Zhang[3], Guangyao Shan[1], Zhengyang Hu[1], Zhencong Chen[1], Zongwu Lin[1], Wei Jiang[1], Qun Wang[1], Junjie Xi[1*], Shanye Yin[2*], Cheng Zhan[1*]

[1]Department of Thoracic Surgery, Zhongshan Hospital, Fudan University, Shanghai, China; [2]Department of Pathology, Albert Einstein College of Medicine, Bronx, United States; [3]Department of Thoracic Surgery, Sichuan Cancer Hospital, University of Electronic Science and Technology of China, Sichuan, China

*For correspondence:
xi.junjie@zs-hospital.sh.cn (JX);
Shanye.yin@einsteinmed.edu
(SY);
czhan10@fudan.edu.cn (CZ)

†These authors contributed
equally to this work

Competing interest: The authors
declare that no competing
interests exist.

Reviewing Editor: Tony Ng,
King's College London, United
Kingdom

## eLife Assessment

This study reports single-cell RNA sequencing results of lung adenocarcinoma, comparing four treatment-naive and five post-neoadjuvant chemotherapy tumor samples. Of interest is the delineation of two macrophage subtypes: Anti-mac cells (CD45+CD11b+CD86+) and Pro-mac cells (CD45+CD11b+ARG+), with the proportion of Pro-mac/pro-tumorigenic cells significantly increasing in LUAD tissues after neoadjuvant chemotherapy. In terms of significance, the findings might be **useful**. However, issues remain after the revision with lengthy descriptive clustering-type analysis, insufficient statistical support, and inefficient figure presentation. As it stands, the level of supportive evidence is **inadequate**.

**Abstract** Chemotherapy is widely used to treat lung adenocarcinoma (LUAD) patients comprehensively. Considering the limitations of chemotherapy due to drug resistance and other issues, it is crucial to explore the impact of chemotherapy and immunotherapy on these aspects. In this study, tumor samples from nine LUAD patients, of which four only received surgery and five received neoadjuvant chemotherapy, were subjected to scRNA-seq analysis. In vitro and in vivo assays, including flow cytometry, immunofluorescence, Seahorse assay, and tumor xenograft models, were carried out to validate our findings. A total of 83,622 cells were enrolled for subsequent analyses. The composition of cell types exhibited high heterogeneity across different groups. Functional enrichment analysis revealed that chemotherapy drove significant metabolic reprogramming in tumor cells and macrophages. We identified two subtypes of macrophages: Anti-mac cells (CD45+CD11b+CD86+) and Pro-mac cells (CD45+CD11b+ARG +) and sorted them by flow cytometry. The proportion of Pro-mac cells in LUAD tissues increased significantly after neoadjuvant chemotherapy. Pro-mac cells promote tumor growth and angiogenesis and also suppress tumor immunity. Moreover, by analyzing the remodeling of T and B cells induced by neoadjuvant therapy, we noted that chemotherapy ignited a relatively more robust immune cytotoxic response toward tumor cells. Our study demonstrates that chemotherapy induces metabolic reprogramming within the tumor microenvironment of LUAD, particularly affecting the function and composition of immune cells such as macrophages and T cells. We believe our findings will offer insight into the mechanisms of drug resistance and provide novel therapeutic targets for LUAD in the future.

## Introduction

Lung cancer is the most common cancer among all human tumor types, with more than $1.7 \times 10^6$ new cases worldwide each year. According to the Global Cancer Report data, lung adenocarcinoma (LUAD) accounts for most lung cancers (*Siegel et al., 2020*). The application of adjuvant or neoadjuvant chemotherapy (NCT) has significantly improved the long-term survival of LUAD patients. At present, for most LUADs that need chemotherapy after being assessed, chemotherapy will be used before and after surgery (*Ren et al., 2021*). However, chemotherapy drugs are highly toxic and can often become ineffective (*Ekoh et al., 2022*). In addition, continued ineffective chemotherapy will lead to the generation of drug-resistant tumor cell clones (*Cai et al., 2020*; *Bondarenko et al., 2021*) and a delay in tumor removal. Almost all cancer patients show inherent or acquired drug resistance, leading to treatment failure and unsatisfactory overall survival. Therefore, to accurately develop therapies that can overcome drug resistance, it is essential to understand the alterations in the tumor microenvironment (TME) driven by chemotherapy.

Many studies have increasingly proved the TME to be an essential source of intratumoral heterogeneity (*Katsumata et al., 2018*). The heterogeneity within the TME encompasses not only the variations between different tumor cells but also among various stromal and immune cell types. Investigating the dynamic changes in multiple cell populations within the TME of LUAD following chemotherapy may provide crucial insights into overcoming chemotherapy resistance in LUAD. In this study, we demonstrated the changes in the microenvironment of LUAD with chemotherapy. In particular, we focused on the effect of chemotherapy on the metabolic reprogramming of tumor cells, stromal cells, and immune cells.

Formerly, it was generally believed that consuming glucose in TME by cancer cells may promote nutritional competition, a metabolic mechanism of immunosuppression (*Ho et al., 2015*). However, recent studies have shown that tumor-infiltrating immune cells rely on glucose for their energy needs and functionality, with immune cells, particularly macrophages, consuming more glucose than malignant cells. The impaired immune cell metabolism in the TME helps tumor cells escape immunity (*Chang et al., 2015*). The internal metabolic changes in the cells drive immune cells and cancer cells to preferentially obtain glucose and glutamine. It is believed that the selective cellular allocation of these nutrients can be used to develop therapeutic and imaging strategies to enhance or monitor the metabolic processes and activities of specific cell populations in TME (*Reinfeld et al., 2021*). Metabolic reprogramming in various cell types in the TME after undergoing chemotherapy may be an essential feature that affects chemotherapy. Our research fully demonstrated the metabolic reprogramming landscape of tumor cells, stromal cells, and immune cells before and after chemotherapy.

## Results

### Single-cell transcriptomic profiling of LUAD

A total of nine patients with non-metastatic LUAD underwent lobectomy with curative intent in the Department of Thoracic Surgery, Zhongshan Hospital of Fudan University. Among them, five received three cycles of preoperative neoadjuvant combination chemotherapy with cisplatin plus pemetrexed (defined as NCT group), while others only received surgery (defined as the control group). Following resection, a malignant lung tumor sample was obtained from each patient, rapidly digested to a single-cell suspension, and analyzed using 10×scRNA-seq (*Figure 1a*). After quality control, a total of 83,622 cells that met the inclusion criteria were subjected to subsequent analyses, with 33,567 and 50,055 cells derived from the control and NCT groups, respectively (*Figure 1a–c*, *Figure 1—figure supplement 1a*). Next, we classified cell types through dimensional reduction and unsupervised clustering using the Seurat package and relative maker genes.

Using the SingleR package, the CellMarker dataset, and our previous studies (*Chen et al., 2021b*; *Chen et al., 2021a*), we identified cell clusters that could be assigned to known cell lineages: epithelial cells (marked by SFTA2 and KRT8), T cells (marked by CD3D and TRBC2), B cells (marked by CD79A and CD19), endothelial cells (ECs, marked by EMCN and CXorf36), mast cells (marked by TPSB2 and TPSAB1), macrophages (marked by CD68 and APOE), monocytes (marked by FGL2 and LGALS2), fibroblasts (marked by LUM and DCN), and neutrophils (marked by FCGR3B and CMTM2). Meanwhile, the consensus clustering of these cells also exhibited the consistency and homogeneity of the expression profile within each identified cell type (*Figure 1d*). For instance, clusters 1, 3, 6, 7,

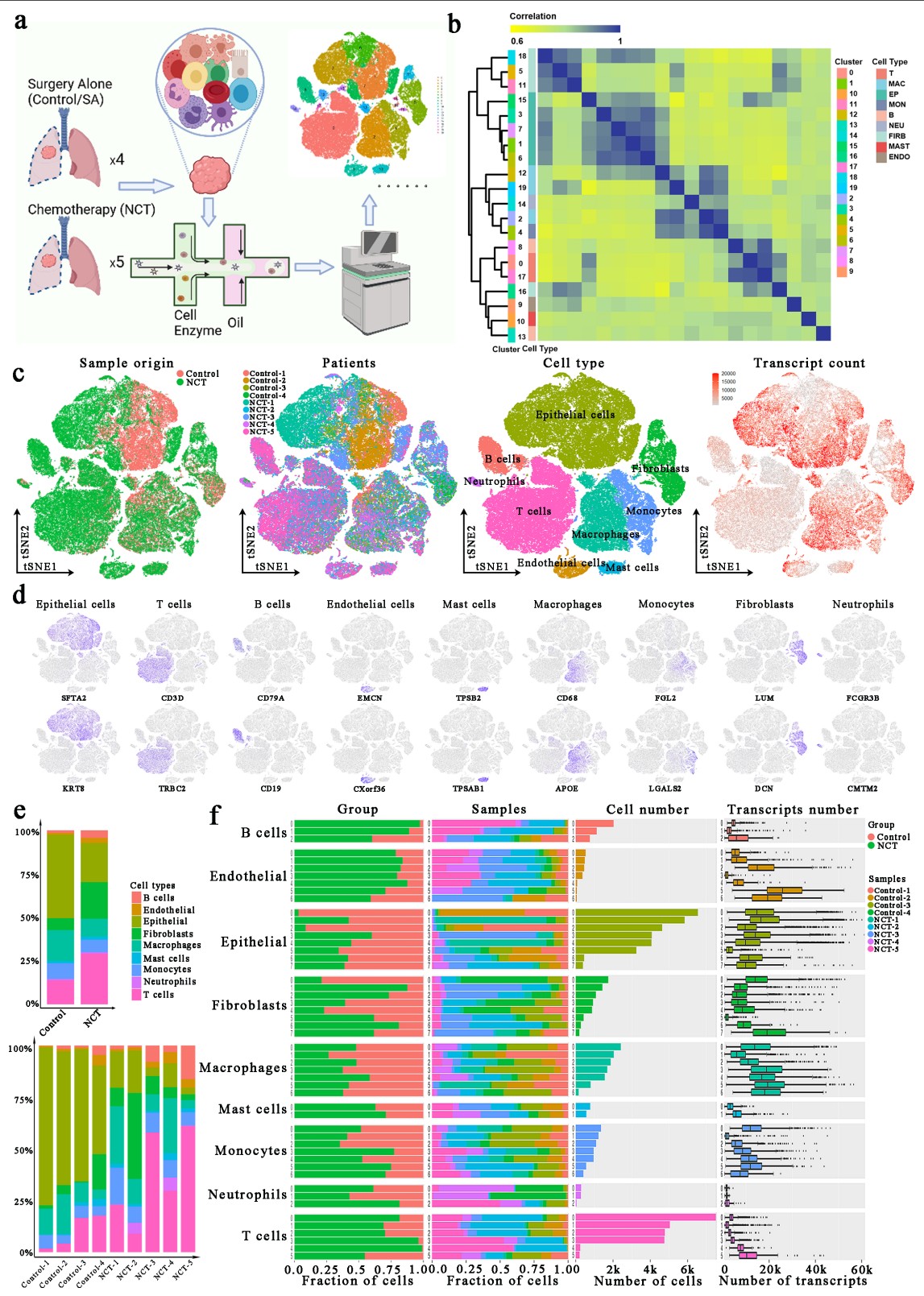

**Figure 1.** Single-cell atlas of lung adenocarcinoma (LUAD) tissues from the control and neoadjuvant chemotherapy (NCT) group. (**a**) Workflow depicting collection and processing of LUAD samples for scRNA-seq analysis. (**b**) Consensus clustering based on the correlations among the 20 clusters identified through the *t*-distributed random neighborhood embedding (TSNE) algorithm. (**c**) tSNE of the 83,622 cells enrolled here, with each cell color indicating: its sample type of origin, the corresponding patient, predicted cell type, and the transcript counts. (**d**) Expression of marker genes for the cell types

*Figure 1 continued on next page*

*Figure 1 continued*

defined above each panel. (**e**) The proportion of each cell type in different groups and samples. (**f**) For each of the 8 epithelial subclusters and 43 non-epithelial clusters (left to right): the fraction of cells originating from the three groups, the fraction of cells originating from each of the nine patients, the number of cells and box plots of the number of transcripts (with plot center, box, and whiskers corresponding to the median, IQR and 1.5× IQR, respectively).

The online version of this article includes the following figure supplement(s) for figure 1:

**Figure supplement 1.** tSNE clusters and marker genes across cell types.

and 15, all designated as epithelial cells, were adjacent to each other in the consensus heatmap. This result confirms the robustness and reliability of our data preprocessing. Detailed distributions of these marker genes in each cluster are depicted in *Figure 1—figure supplement 1b*.

By comparing the composition of different types of cells in each group, we noticed TME heterogeneity: the proportion of cells other than tumor cells, especially immune cells (mainly T and B), was significantly higher in the NCT (*Figure 1e*). Therefore, to identify subclusters within each of these nine major cell types, we observed a complex cellular ecosystem containing 8 different epithelial subclusters and 43 non-epithelial clusters. Interestingly, the epithelial subclusters, mainly composed of cancer cells, were highly patient-specific, while the immune cell subclusters mostly consisted of cells derived from four or more patients (*Figure 1f*). This observation demonstrated the substantial variation and heterogeneity of TME among groups and individuals. Therefore, we further explored these alterations associated with the therapeutic regimen in greater detail for the primary cell types in subsequent analyses.

## Metabolic reprogramming in LUAD driven by NCT

Metabolic reprogramming is a hallmark of malignant tumors. Recent studies have also shown that tumors' metabolic characteristics and preferences change during cancer progression (*Faubert et al., 2020*). In each type of cell derived from the control and NCT groups, more significantly upregulated metabolic pathways were enriched in cancer cells, nonmalignant epithelial cells, fibroblasts, and macrophages (*Figure 2a*). The enrichment of oxidative phosphorylation (OXPHOS), glycolysis, pyruvate metabolism, and the tricarboxylic acid cycle indicates active glucose metabolism in these four cell types. By analyzing the activity of metabolic pathways in cells from different sources, we found that the activity scores of the metabolic pathways of tumor cells and macrophages were significantly higher than those of other types of cells. Notably, the metabolic pathway activity of macrophages and malignant cells increased after chemotherapy (*Figure 2b*).

## Changes in metabolism and gene expression of tumor cells after NCT

To accurately analyze the effect of chemotherapy on the cancer cells, we first reclustered the epithelial cells, and 12 clusters were identified (*Figure 3a*). Copy number variations (CNVs) (*Figure 3—figure supplement 1a and c*) and marker genes were used to accurately separate malignant and nonmalignant epithelial cells in control and NCT samples. They were finally defined as MalignantSA cells (marker genes: FOXL2/MET/CD74), MalignantNCT cells (marker genes: RAC1/MAF/CXCL1), and nonmalignant cells (marker genes: ABCA3/SFTPB/LPCAT) (*Figure 3d*). These marker genes were further confirmed by immunofluorescence experiments (*Figure 3—figure supplement 1d*). We found that the proportion of malignant cells was significantly reduced after chemotherapy (*Figure 3b and c*, *Figure 3—figure supplement 1b*). Although malignant cells were significantly reduced after chemotherapy, genetic aberrations by CNVs analysis revealed that MalignantNCT cells exhibited significantly higher malignant scores compared to MalignantSA cells (*Figure 3—figure supplement 1a*).

We performed trajectory analysis to track the reprogramming of epithelial cells across the three groups. Nonmalignant cells evolved in two directions and developed into two clusters of cells (*Figure 3e*). In this evolutionary process, glycolysis-related genes (ENO1, LDHB, GAPDH), OXPHOS-related genes (NDUFA4), mitochondrial repair-related genes (TOMM7), glucose and lipid metabolism regulation genes (S100A16), ATPase activity-related genes (CCT6A), tumor immune regulation-related genes (CCL20, CXCL1, PAEP, PPP1R14B), hypoxia response regulation genes (CHCHD2), apoptosis regulation genes (MEG3, CEACAM5), mRNA alternative splicing-related genes (LSM5), and Ras-related protein (RAC1, RALA) gradually increased over time in the pseudotime analysis. These findings indicate that these genes play an essential role in the transformation of epithelial cells into tumor

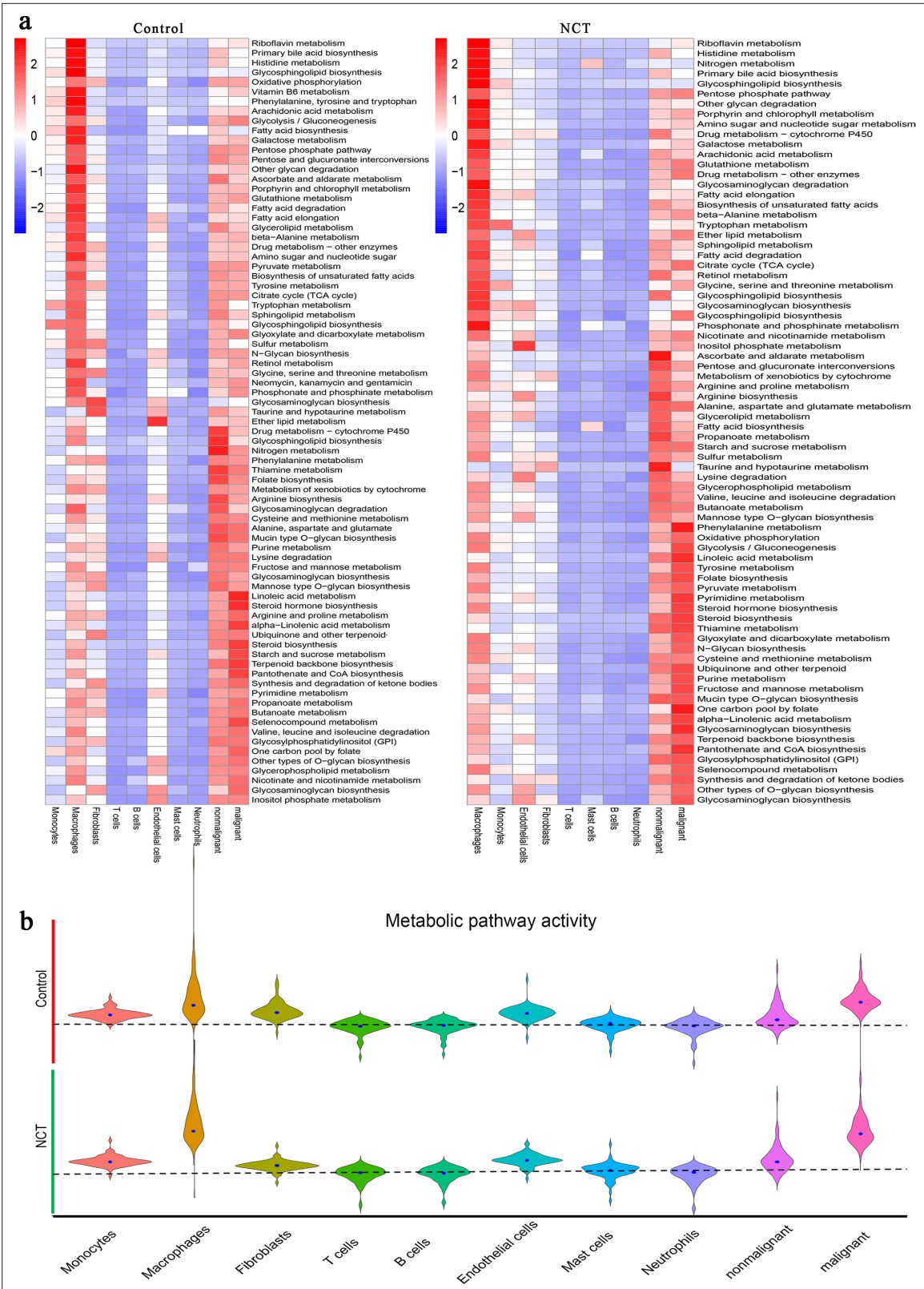

**Figure 2.** Metabolic reprogramming in lung adenocarcinoma driven by neoadjuvant chemotherapy (NCT). (**a**) The metabolic pathway activities of different cells from the control group and NCT group showed significant differences. (**b**) The metabolic pathway activity of macrophages and malignant cells increased significantly after chemotherapy.

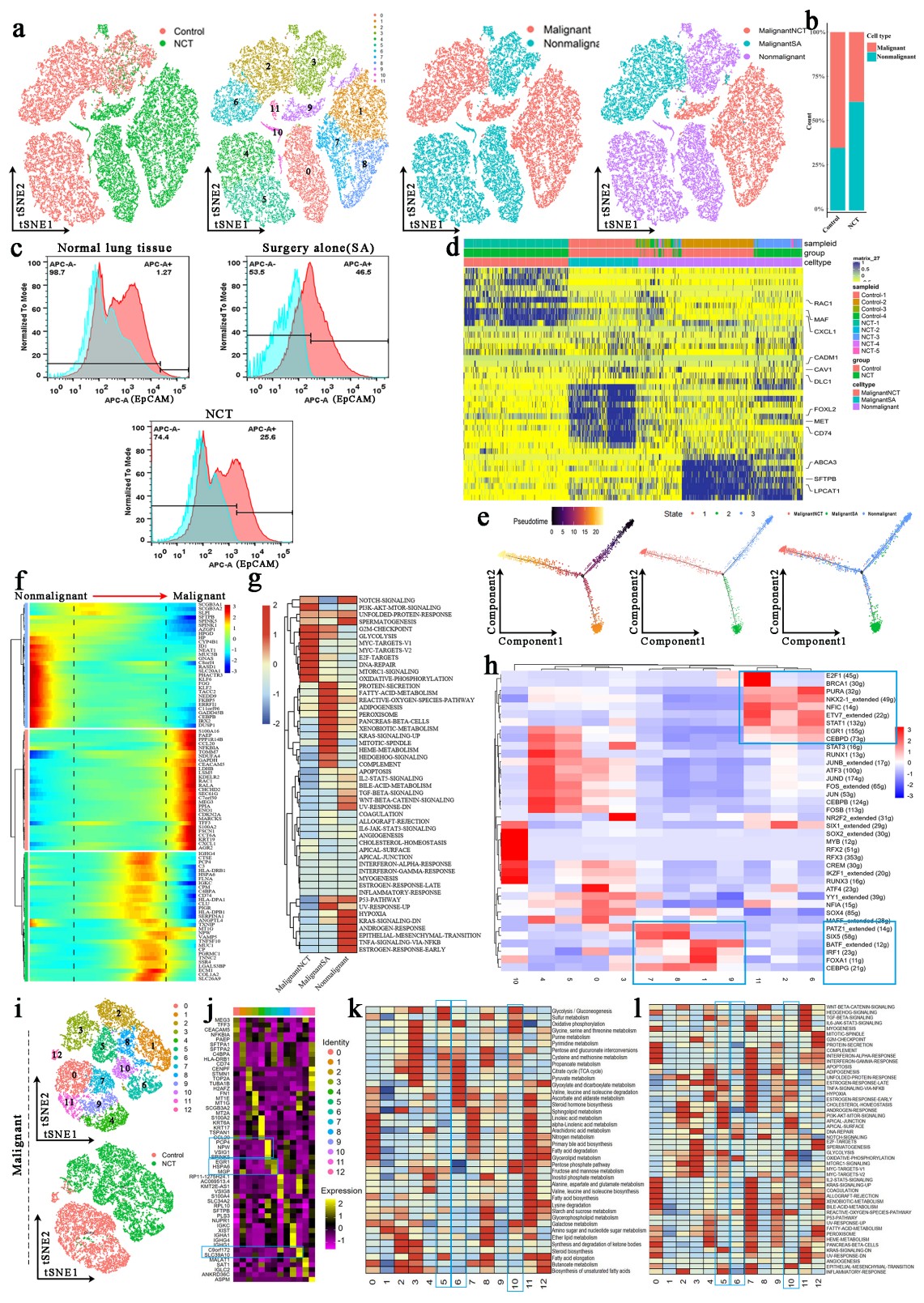

**Figure 3.** Tumor cells and epithelial cells had significant phenotypic changes before and after chemotherapy. (**a**) The *t*-distributed random neighborhood embedding (TSNE) plots and overview of the tumor cells and epithelial cells. (**b**) The proportion of malignant cells and nonmalignant cells in the control group and neoadjuvant chemotherapy (NCT) group. (**c**) Flow cytometry showed that the proportion of malignant cells was significantly reduced after chemotherapy and immunotherapy. (**d**) Marker genes of MalignantSA cells, MalignantNCT cells, and nonmalignant cells. (**e**)

*Figure 3 continued on next page*

*Figure 3 continued*

Pseudotime analysis showed that nonmalignant cells evolved in two directions. (**f**) The heatmap shows that a series of genes play an important role in transforming epithelial cells into tumor cells. (**g**) Gene Set Variation Analysis (GSVA) was performed for malignant and nonmalignant cells. (**h**) Single-Cell Regulatory Network Inference and Clustering (SCENIC) analysis revealed the hub genes in the malignant transformation of epithelial cells. (**i**) The tSNE plots for reclustered malignant cells. (**j**) Marker genes of 13 subclusters from malignant cells. (**k**) Metabolic characteristics in different malignant cell subclusters. (**l**) GSVA reveals the characteristics of pathway activity in different malignant cell subclusters.

The online version of this article includes the following figure supplement(s) for figure 3:

**Figure supplement 1.** CNVs, gene expression, and pathway activity of malignant and nonmalignant epithelial cells.

cells (*Figure 3f*, *Figure 3—figure supplement 1e*). Correspondingly, during the process of epithelial cells transforming into malignant tumor cells, the activity of the glycolysis pathway, OXPHOS pathway, angiogenesis pathway, DNA repair pathway, and mTORC1 signaling pathway gradually increased over time. However, P53 pathway, apoptosis signaling pathway activity then steadily decreased (*Figure 3—figure supplement 1f*). Similarly, we performed Gene Set Variation Analysis (GSVA) on malignant and nonmalignant cells from the three groups. We found that the glycolysis pathway, OXPHOS pathway, MYC-targets, E2F-targets, DNA repair pathway, and mTORC1 signaling pathway were significantly enriched in MalignantNCT cells derived from the NCT group (*Figure 3g*). The metabolic reprogramming enables cancer cells to resist anticancer drugs, thereby developing chemoresistance (*Tuerhong et al., 2021*). To find the hub genes that cause the malignant transformation of epithelial cells, through Single-Cell Regulatory Network Inference and Clustering (SCENIC) analysis, we found that E2F1, BRCA1, PURA, NKX2-1, NFIC, ETV7, STAT1, EGR1, and CEBPD were highly expressed in malignant cells (cluster 2, 6, 11) from the control group. In contrast, the malignant cells from the NCT group (clusters 1, 7, 8, 9) have a high expression of transcription factors (TFs) such as PATZ1, SIX5, BATF, IRF1, FOXA1, and CEBPG. After NCT, the increased expression of these TFs promoted the occurrence of LUAD complex phenotypic remodeling (*Figure 3h*).

Tumor cells have significant heterogeneity. We reclustered the malignant cells and obtained 13 subclusters (*Figure 3i and j*). Clusters 1, 2, 5, 6, 8, and 10 were derived from the NCT group. Through the analysis of the metabolism of these cell subclusters, we found that clusters 5 (marker genes: PCP4/NPW/VSIG1), 6 (marker genes: ERG1/HSPA6), and 10 (marker genes: C9orf172/SLC39A10) from the NCT group showed high levels of glycolysis, OXPHOS, and pyruvate metabolism (*Figure 3k*). GSVA also showed that the glycolysis and OXPHOS signaling pathway-related genes were significantly enriched in clusters 5, 6, and 10 (*Figure 3l*). Similarly, we reclustered nonmalignant cells to obtain 16 subclusters, of which clusters 1, 3, 8, 11, 12, 13, and 15 were from the NCT group, and the rest were from the control group (*Figure 3—figure supplement 1g and h*). Clusters 4 and 7 from the control group showed high levels of glycolysis and OXPHOS (*Figure 3—figure supplement 1i and j*), which contrasts with the glucose metabolism observed in malignant cells from the control group.

## Changes in stroma cells resulted from NCT

To investigate stromal cell dynamics in the TME, we obtained 8944 presumed stromal cells, as shown in *Figure 1c*. We reclustered them into five subpopulations, including COL14A1-positive fibroblasts, endothelial-1, endothelial-2, myofibroblasts, pericytes, and smooth muscle cells (SMCs) (*Figure 4a–c*; *Kim et al., 2020*; *Xie et al., 2018*; *Vanlandewijck et al., 2018*). Detailed expression of the marker genes in each cell type is outlined in *Figure 4c*. Herein, we noticed a significant difference between the distribution of each of these five clusters in patients receiving varied types of treatment. The COL14A1-positive fibroblasts comprised the main fibroblast types in NCT groups, in which both endothelial 1 and 2 were mainly found. Pericyte and SMCs were presented in all three groups. In contrast, myofibroblasts exclusively originated from the control group. According to previous research, myofibroblasts have been described as cancer-associated fibroblasts that participate in extensive tissue remodeling, angiogenesis, and tumor progression (*Kim et al., 2020*; *Xie et al., 2018*). Therefore, this finding revealed that NCT and immunotherapy significantly altered the stromal cell composition in the TME.

To explore the activity of known biological pathways in these stromal cells, we performed functional enrichment analysis. In particular, GSVA exhibited that endothelial 1 and 2 shared several upregulated pathways related to cell proliferation and fate regulation, including IL6-JAK-STAT3, TGFβ, and WNT-β catenin signaling. Besides, pathways associated with energy metabolisms such

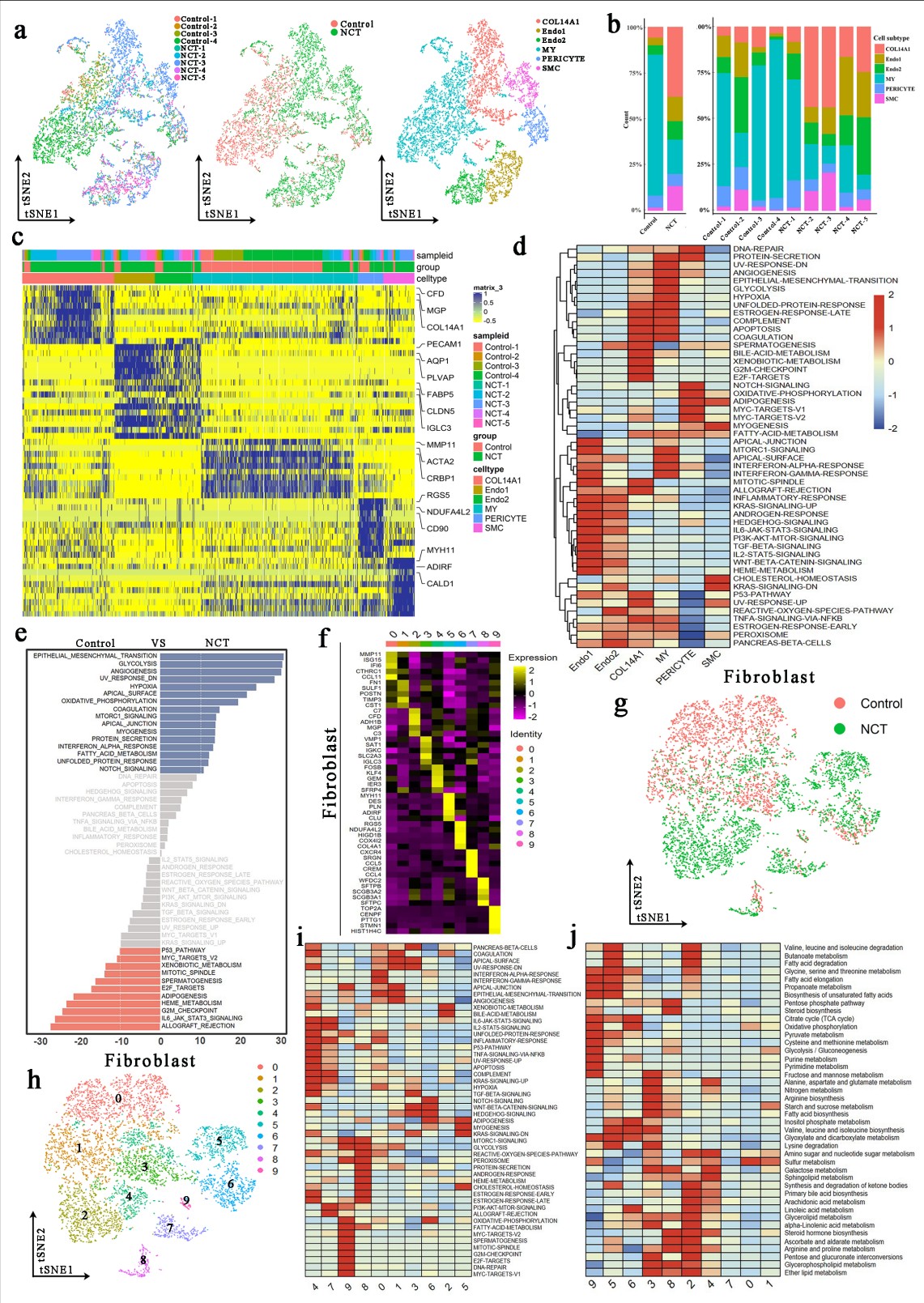

**Figure 4.** The scRNA profile of stromal cells derived from lung adenocarcinoma (LUAD) samples in the control and neoadjuvant chemotherapy (NCT) groups. (**a**) The *t*-distributed random neighborhood embedding (TSNE) plots an overview of the six clusters of stromal cells. (**b**) Proportions of the six predicted clusters of stromal cells in different groups and samples. (**c**) Heatmap exhibiting the expression level of marker genes in each stromal cell cluster. (**d**) Gene Set Variation Analysis (GSVA) estimated the pathway activation levels of different stromal cell subtypes. The scores have been

*Figure 4 continued on next page*

*Figure 4 continued*

normalized. (**e**) GSVA revealed the activation level of hallmark pathways in stromal cells (control vs. NCT groups) (**f**) Heatmap exhibiting the expression level of marker genes in each fibroblast cluster. (**g, h**) The tSNE plots reveal the group origins (**g**) and predicted subclusters (**h**) of fibroblast. (**i, j**) GSVA estimated the pathway activation levels of different fibroblast subtypes.

The online version of this article includes the following figure supplement(s) for figure 4:

**Figure supplement 1.** Metabolic and transcriptional analysis of fibroblasts.

as glycolysis and hypoxia were upregulated in myofibroblast, whereas pericyte was characterized by enriched OXPHOS and adipogenesis (*Figure 4d*). Meanwhile, when comparing the GSVA scores of these biological processes between patients from the control or NCT groups, we noted that the stromal cells exhibited enhanced metabolic levels after NCT, as represented by upregulated glycolysis, OXPHOS, and fatty acid metabolism pathways (*Figure 4e*).

Considering the essential role of fibroblasts and their complicated function in shaping the TME, we further reclustered them into 10 subgroups (*Figure 4f–h*). As shown in *Figure 4i and j*, the GSVA score of the metabolic pathways, including glycolysis and OXPHOS, and pyruvate metabolism and citrate cycle (TCA cycle), were upregulated in clusters 5, 6, and 9. Intriguingly, the upregulation of these pathways was mainly observed in the NCT groups (*Figure 4—figure supplement 1a and b*). The three clusters were represented by distinct gene expression profiles, such as overexpressed MYH11 in cluster 5, RGS5 in cluster 6, and TOP2A in cluster 9. Since the potential involvement of these genes in the manipulation of fibroblast metabolism has never been proposed, they might serve as new specific markers of the fibroblast subtype with such a high metabolic rate in the TME. Besides, the SCENIC analysis demonstrated that MEF2C, NFIA, and RAD21 might drive the formation of these clusters, respectively (*Figure 4—figure supplement 1c*). Further in vitro studies are required to elucidate these notable fibroblasts' potential function and driver genes in LUAD's development and response to NCT. Conclusively, cellular dynamics in stromal cells support a consistent phenotypic shift of fibroblasts toward an increased metabolic level after preoperative chemotherapy.

## Chemotherapy drove tumor-associated macrophages to turn more into phenotypes that promote tumor progression

In the process of cancer formation, tumor-associated macrophages (TAMs) have an essential influence on the inflammatory response in the TME (*Malfitano et al., 2020*). To study the effects of chemotherapy on TAMs, we first extracted all macrophages (10,526 cells) and reclustered them into 10 cell clusters (*Figure 5a*). From *Figure 1e*, we can see that the proportion of macrophages after chemotherapy was reduced.

The cell clusters derived from the control group were 1/2/3/5/7, those from the NCT group were mainly 0/4/8, and the number of cells in the 6/9 clusters from the control group and the NCT group was similar (*Figure 5a*). The proportion of cells in cluster 0 (marker genes: CXCL8/CCL20/CHIT1), 4 (marker genes: CCL3/CCL4/SEPP1), and 8 (marker genes: ARG2/S100A2) decreased after chemotherapy, while the remaining cell clusters increased (*Figure 5b and c*). Through the GSVA, we found that glycolysis, angiogenesis, PI3K-AKT-mTOR-signaling, IL6-JAK-STAT3-signaling, hypoxia, TGF-β-signaling, and other signaling pathways were significantly enriched in clusters 0/1/8. Promoting inflammation-related signaling pathways such as TNF signaling via NFKB, inflammatory response, Notch signaling, fatty acid metabolism, and oxidative phosphorylation were increased dramatically in clusters 2/4/7/9 (*Figure 5d*).

Similarly, we found that glycolysis/gluconeogenesis, amino sugar and nucleotide sugar metabolism, alanine, aspartate, and glutamate metabolism were more active in the clusters 0/1/8. In contrast, OXPHOS, citrate cycle, pyruvate metabolism, fatty acid elongation, fatty acid biosynthesis, etc., were more active in clusters 2/4/7/9 (*Figure 5e*). According to the GSVA, the glycolysis/gluconeogenesis signaling pathway was significantly enriched in macrophages from the NCT group. In contrast, macrophages from the control group showed a high activity in OXPHOS, fatty acid elongation, fatty acid degradation, fatty acid biosynthesis, and citrate cycle (TCA cycle) (*Figure 5f*). These results indicate that significant metabolic reprogramming occurred in TAMs after chemotherapy, and different TAMs cell clusters also showed huge metabolic differences. In general, our results revealed that chemotherapy could promote glycolysis of TAMs and inhibit fatty acid metabolism.

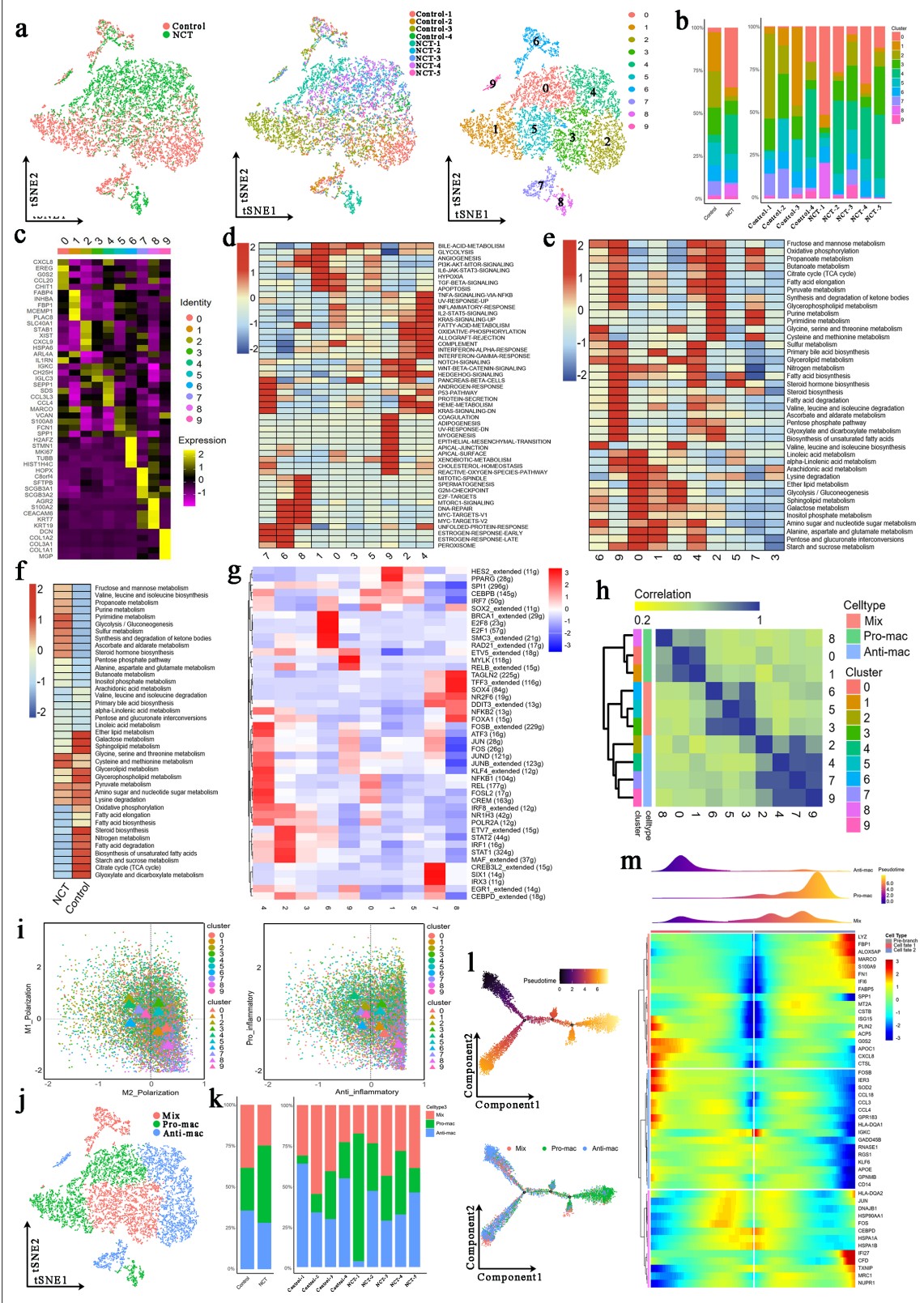

**Figure 5.** Three newly identified subtypes of tumor-associated macrophages (TAMs) display distinct genetic and metabolic features. (**a**) The *t*-distributed random neighborhood embedding (TSNE) plots show the group origins, sample origins, and clusters of TAMs. (**b**) Proportions of the 10 clusters of TAMs in different groups and samples. (**c**) Marker genes of the 10 clusters of TAMs. (**d**) Gene Set Variation Analysis (GSVA) was performed for the 10 clusters of TAMs. (**e**) The activity of various metabolic processes in the 10 clusters of TAMs. (**f**) GSVA was performed for TAMs from the control

*Figure 5 continued on next page*

*Figure 5 continued*

group and neoadjuvant chemotherapy (NCT) group. (**g**) Single-Cell Regulatory Network Inference and Clustering (SCENIC) analysis was performed for the 10 clusters of TAMs. (**h**) Consensus clustering based on the correlations among the 10 clusters of TAMs identified through the tSNE algorithm. (**i**) Polarization score (left) and inflammatory score (right) for 10 clusters of TAMs based on the expression of polarization marker genes and inflammatory genes. (**j**) The tSNE plots for three types of TAMs. (**k**) Proportions of the three subtypes of TAMs in different groups and samples. (**l**) Development trajectory analysis for the three subtypes of TAMs. (**m**) Pseudotime analysis revealed a series of genes that affect the differentiation and development of macrophages.

To explore the key genes that regulate the differences in the metabolism of each subcluster of macrophages, we performed a SCENIC analysis. We found that HES, PPARG, SPI1, CEBPB, and IRF7 were highly expressed in cluster 0/1, which may be the key genes that regulate the conversion of macrophages into M2-like TAMs, while clusters 2/4/7/9 highly expressed STAT1, STAT2, NFKB1, JUN, and FOS that regulate the conversion of macrophages to M1-like TAMs (*Figure 5g*).

According to the gene expression of macrophages, we divided these 10 clusters of cells into three subtypes of macrophages through cluster analysis (*Figure 5h*). We scored the expression levels of pro-inflammatory and anti-inflammatory genes in all macrophages. We displayed each color-coded macrophage subtype's M1 and M2 scores (left) and pro-inflammatory and anti-inflammatory scores (right) through a scatter plot.

Similarly, we found that 0/1/8 cluster cells exhibited M2-like polarization and anti-inflammatory properties, while 2/4/7/9 exhibited M1-like polarization and pro-inflammatory properties (*Figure 5i*). Based on these analyses, we divided these 10 clusters of macrophage subtypes into three categories: M1-like polarized phenotype was defined as Anti-mac; M2-like polarized phenotype was defined as Pro-mac; and those without obvious polarized phenotype were defined as Mix (*Figure 5j*). We found that the proportion of Pro-mac in the TME increased after chemotherapy, especially in the case of NCT-1 (*Figure 5k*). Interestingly, via trajectory analysis we found that two subtypes, Anti-mac and Mix, can be converted to Pro-mac. In this evolution process, the high expression of LYZ, FBP1, ALOX5AP, MARCO, S100A9, FN1, CXCL8, APOC1CTSL, and other genes may have played an essential role in promoting the conversion of Anti-mac to Pro-mac (*Figure 5l and m*). This suggests that we can change the phenotype of TAMs in the TME by altering the expression of these genes.

## Chemo-driven Pro-mac and Anti-mac metabolic reprogramming exerted diametrically opposite effects on tumor cells

To further verify the remodeling effect of chemotherapy on the functional phenotype of TAMs in the TME, we first used the FindAllMarkers function in the Seurat package to find the marker genes of Pro-mac, Anti-mac, and Mix cells. Pro-mac was mainly characterized by high expression of CXCL8, ARG1, CREM, CD206, STAT6, CCL22, MMP7, and CCL3L3, while Anti-mac was mainly characterized by high expression of CD86, HLA-DR, PLAC8, CXCL10, COX2, IL15R, and SCGB3A1 (*Figure 6a*). Based on these marker genes, we sorted out Anti-mac cells (CD45+CD11b+CD86+) and Pro-mac cells (CD45+CD11b+ARG +) by flow cytometry (*Figure 6b*). To verify whether the cells we sorted were the cell population we wanted, we reverified the positive rates of Pro-mac and Anti-mac cells by flow cytometry (*Figure 6c*). Our results showed that the proportion of Pro-mac cells in LUAD tissues after NCT increased significantly (*Figure 6d*). In fact, by performing immunofluorescence staining on LUAD tissue samples derived from surgery alone and NCT, we also found that the proportion of cells marked by the marker gene CD206 of M2-like TAMs increased significantly after chemotherapy (*Figure 6e*). Macrophages can promote tumor progression by secreting many cytokines. By analyzing the differentially expressed genes of Pro-mac and Anti-mac cells, we found IL10, PDCD1LG2, PDGF, VEGF, MMP9, CXCL9, CXCR4, IL22, KLF4, and TGF-β were highly expressed in Pro-mac cells that promote tumor growth, angiogenesis, and suppress tumor immunity (*Figure 6f*). We obtained the Pro-mac and Anti-mac cells from 12 cases (six cases of surgery alone, six cases of surgical samples after NCT) by flow cytometry. We named them Control Anti-mac, Control Pro-mac, NCT Anti-mac, and NCT Pro-mac. After placing them in a cell culture flask for 24 hr, the content of some key cytokines in the supernatant of the culture medium was detected by ELISA. The levels of MMP9, EGF, and VEGF secreted by Pro-mac after NCT were significantly higher than those of Pro-mac from the surgery alone group. MMP9, EGF, VEGF, and IL10 secreted by Pro-mac were significantly higher than Anti-mac (*Figure 6g*). Similarly, when Control Anti-mac, Control Pro-mac, NCT Anti-mac, and NCT Pro-mac were inoculated

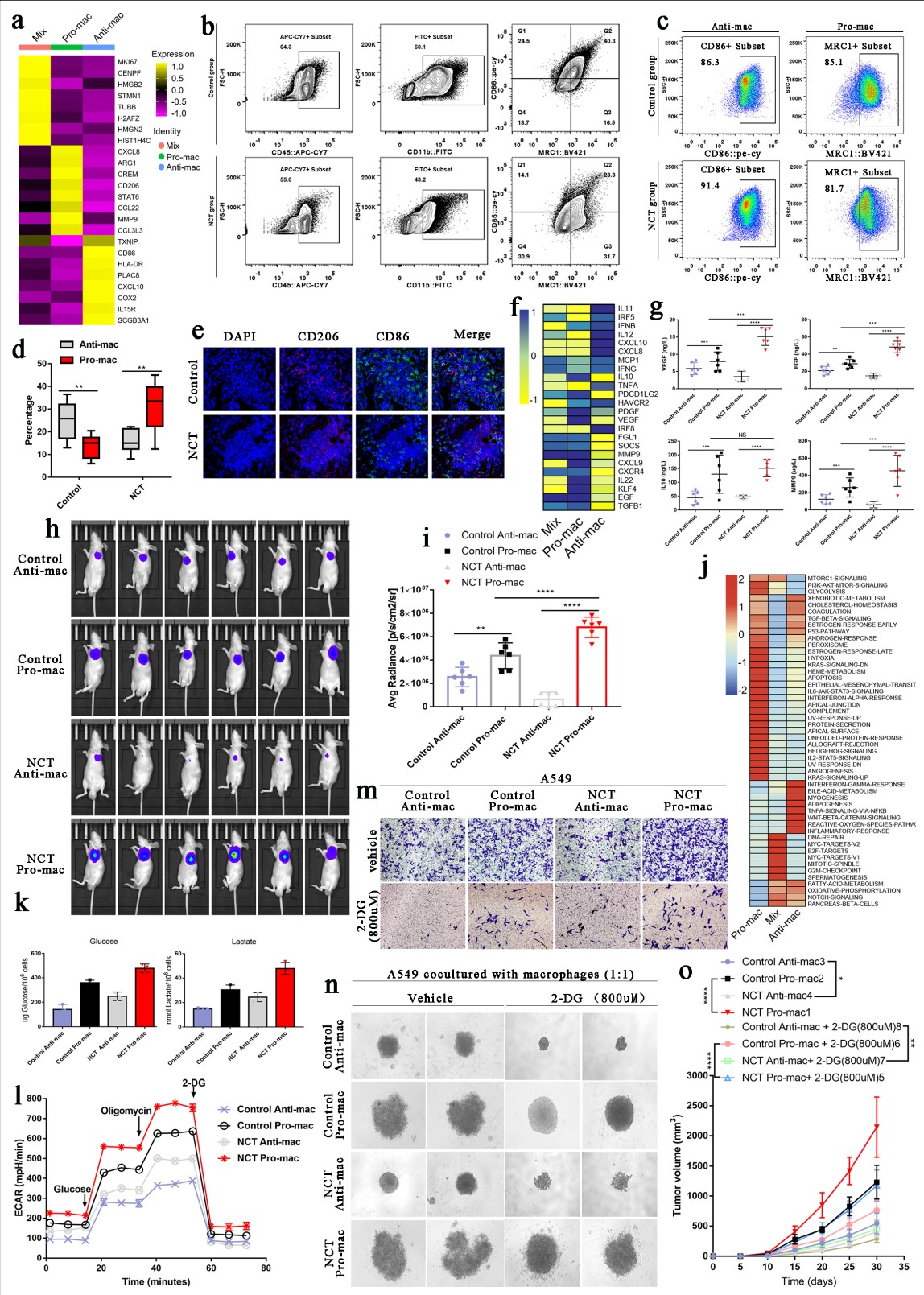

**Figure 6.** Metabolic switching in tumor-associated macrophages (TAMs) contributed to diametrical effects on tumor cells. (**a**) The heatmap shows the essential marker genes for three subtypes of TAMs. (**b**) Based on Pro-mac and Anti-mac marker genes, these two types of cells were sorted by flow cytometry from lung adenocarcinoma tissue. (**c**) Flow cytometry verified the sorted cells. (**d**) The proportion changes of Pro-mac and Anti-mac cells in lung adenocarcinoma tissues before and after chemotherapy. (**e**) Immunofluorescence showed the changes in the proportion of TAMs with high CD206

*Figure 6 continued on next page*

*Figure 6 continued*

and CD86 after neoadjuvant therapy. (**f**) The heatmap shows the differences in the cytokines secreted by the three subtypes of macrophages. (**g**) ELISA detected the secretion of VEGF, EGF, IL10, and MMP9. (**h**) The intensity of fluorescence changes in Luciferase-labeled A549 cells mixed with different TAMs. (**i**) The histogram shows the average fluorescence intensity emitted by the subcutaneous tumor. (**j**) Gene Set Variation Analysis (GSVA) performed for Pro-mac, Anti-mac, and Mix. (**k**) Glucose uptake and lactate production in the TAMs cell subtypes. (**l**) Seahorse XFe96 cell outflow analyzer detected the glycolysis level of TAMs cell subtypes (extracellular acidification rate: ECAR). (**m**) Transwell experiment detected the influence of TAMs subtypes on the invasion ability of A549 cells. (**n**) The 3D cell culture experiment detected the effect of 2-DG on the spheroidization ability of A549 cells when cultured with subtypes of TAMs. (**o**) In vivo experiments verified the effect of 2-DG on the tumorigenesis ability of A549 after inhibiting glycolysis of TAMs. All error bars are mean ± SD. NS, not significant. ***$p<0.001$, **$p<0.01$, *$p<0.05$; determined by two-tailed Student's *t*-test (95% CI).

The online version of this article includes the following figure supplement(s) for figure 6:

**Figure supplement 1.** The scRNA profile of T cells derived from lung adenocarcinoma (LUAD) samples in the control, neoadjuvant chemotherapy, and immunotherapy groups.

**Figure supplement 2.** The scRNA profile of B cells derived from lung adenocarcinoma (LUAD) samples in the control, neoadjuvant chemotherapy, and immunotherapy groups.

**Figure supplement 3.** Crosstalk between cancer and immune cells.

**Figure supplement 4.** Crosstalk between cancer and immune cells.

subcutaneously with A549 cells at a ratio of 1:1 (reinjection of macrophages 2 weeks later), we also found that NCT Pro-mac can significantly promote tumor growth. Interestingly, NCT Anti-mac in the TME after chemotherapy can significantly inhibit the growth of tumor cells, and this inhibitory ability was stronger than Control Anti-mac (*Figure 6h and i*).

Our previous analysis found that Pro-mac glycolysis-related signaling pathways were significantly enriched, while in Anti-mac, OXPHOS, and fatty acid metabolism signaling pathways were greatly enhanced (*Figure 6j*). In vitro experiments show that NCT Pro-mac's ability to take up glucose and produce lactate was considerably more potent than other cells (*Figure 6k*). It was worth noting that the glycolysis level of NCT Anti-mac was markedly higher than that of Control Anti-mac (*Figure 6l*). When we placed the Pro-mac and Anti-mac in a 24-well plate and co-cultured with A549 cells in the Transwell chamber, we found that NCT Pro-mac can significantly enhance the invasion ability of A549 cells. At the same time, NCT Anti-mac showed a stronger ability to inhibit tumor invasion than Control Anti-mac (*Figure 6m*). However, when we used 2-DG (800 uM, the concentration determined in pre-experiment) to inhibit the glycolysis of TAMs, the ability of Pro-mac to promote tumor progression was significantly weakened, and the power of NCT Anti-mac to suppress tumors was also considerably reduced (*Figure 6m*). By mixing these cells with macrophages for 3D culture, we found that the ability of NCT Anti-mac to inhibit tumor proliferation was significantly weakened when inhibiting its glycolytic activity. This showed that glycolysis could enhance the ability of Pro-mac to promote tumor progression and increase the capacity of Anti-mac to inhibit tumors (*Figure 6n*). Finally, through in vivo experiments, we inoculated a mixture of TAMs and A549 to nude mice and obtained the same experimental results as in *Figure 6m/n* (*Figure 6o*).

## Chemotherapy treatment-induced remodeling of T and B cells

Considering the essential role of the TME, especially the immune infiltration level, in tumor development and response to therapy, we next investigated the characteristics of T and B cells. In our study, 22,530 T cells were detected, which accounted for 26.9% of the total. We noticed that the reclustered T cells could not be visibly distinguished among patients receiving different therapeutic regimens (*Figure 6—figure supplement 1a and b*). According to the expression of a series of canonical markers of T cell subtypes, the T cells were divided into CD4+ T (marked by LTB, CD45RO, etc.), CD8+ T (marked by NKG7, GZMA, GZMB, CD8A, etc.), and Tregs (marked by FOXP3, CTLA4, etc.) (*Lambrechts et al., 2018*; *Chen et al., 2021b*; *Lu et al., 2020*; *Figure 6—figure supplement 1c and d*). The detailed expression profile of these marker genes is exhibited in *Figure 6—figure supplement 1e*. Meanwhile, aside from these previously published T-cell markers, we also noted the specific upregulation of several genes in a particular cluster. At the same time, their expression specificity has not been elucidated yet.

As the major executor of tumor immunology, CD8+ T cells are thought to differentiate into cytotoxic T cells (CTLs) and specifically recognize endogenous antigenic peptides presented by the major histocompatibility complex I, thereby eliminating tumor cells (*Uzhachenko and Shanker, 2019*). By

comparing the composition of T cell subtypes in LUAD cells derived from different groups, we found that the proportion of CD8+ T cells in the NCT group was significantly higher than those in patients receiving only surgical treatment (*Figure 6—figure supplement 1d*). Therefore, we focused on CD8+ T cells for subsequent analyses and reclustered them into five new subgroups, in which clusters 1–4 were mainly derived from the NCT group. In contrast, cluster 5 was predominantly enriched in the control group (*Figure 6—figure supplement 1f–k*).

We next explored the expression profile of genes associated with T cell's function in each CD8+ T subcluster. As depicted in *Figure 6—figure supplement 1i*, clusters 1 and 2 were characterized by upregulated naïve T cell markers, such as TCF7, LEF1, and CCR7, whereas genes associated with immune inhibition, like TIGIT, CTLA4, PDCD1, and HAVCR2, were enriched explicitly in cluster 3. Cytotoxic function-related genes, including GZMA GNLY, PRF1, GZMP, and GZMK, IFNG, IL2, were respectively overexpressed in clusters 4 and 5. Based on this evidence, we defined clusters 1 and 2 as naive T, 3 as regulatory/exhausted T, and 4 and 5 as CTLs. Intriguingly, regarding both the sample origins and expression profiles of CD8+ T cells in clusters 4 and 5, we can reasonably hypothesize that NCT treatment potentially induces the reprogramming of CD8+ cytotoxic cells. To further verify this statement, we performed pseudotime-ordered trajectory analysis to monitor the dynamic view of CD8+ T cells' reprogramming process via Monocle. As shown in *Figure 6—figure supplement 1l–p*, three phases were detected in these clusters. Cluster 1, which exhibited the lowest cytotoxicity, was designated as the 'root' state according to pseudotime.

In contrast, the immune inhibition-related genes like LAG3, TIGIT, and PDCD1, and cytotoxicity-related genes such as GZMB and IFNG were respectively activated in phases 2 and 3. This phenomenon is consistent with our T cell phenotype classification mentioned above. Then, our results showed differentiation paths from naive T to Treg/exhausted cells and cytotoxic cells. Considering the transcriptional changes associated with T cell reprogramming, naive T cells (phase 1) expressing high CCR6 and TCF7 differentiate into two distinct fates, clusters 4 and 5, in phase 3. Notably, the cells positioned at the cluster 4 branch were characterized by higher cytotoxicity than in cluster 5 (*Figure 6—figure supplement 1l, m, and o*). Regarding the sample origins of the two clusters, these findings demonstrated that NCT treatment ignites a relatively more robust immune cytotoxic response toward tumor cells, which could be partly explained by the excessive production of neoantigen caused by NCT-induced DNA damage.

SCENIC analyses suggested that distinct transcriptional mechanisms drove the differentiation of naive T cells to either cluster 4 or 5. As revealed in *Figure 6—figure supplement 1q*, the cytotoxic cells derived from NCT-treated LUAD patients (cluster 4) were characterized by increased activation of FOSL2-extended, REL, YBX1, and NF-KB pathways. In contrast, those from the control group (cluster 5) had upregulated JUN, FOSB, and ELF3 extended pathways. Together, our results revealed that preoperative chemotherapy prompts the naïve T cells to differentiate towards a more cytotoxic phenotype.

As for B cells, only 3902 (4.6%) cells were detected. In total, 475 cells were derived from the control group, while 3427 were from the NCT group (*Figure 6—figure supplement 2a*). Herein, we reclustered the B cells into two subclusters. Based on canonical cell markers, class-switched memory B-cells (marked by CD19, CD37, and HLA-DRA) and plasma cells (marked by IGHA2, IGHG4, and CD38) were defined (*Figure 6—figure supplement 2a–c*). The former compromised the majority of the total B cells (80.7%). Notably, the sample origins of the B cells demonstrated that a higher proportion of plasma cells characterized the control groups. In contrast, the class-switched memory B cells were significantly enriched in preoperatively treated patients.

Meanwhile, we performed GSVA to explore several key biological pathways in the B cells derived from different groups. As depicted in *Figure 6—figure supplement 2d*, B cells from the control group exhibited significant activating ways associated with metabolism and energy supply, including glycolysis and OXPHOS. However, the B cells derived from the NCT group exerted essential roles in most of the pathways, including glycolysis, fatty acid metabolism, apoptosis, and hypoxia. Overall, our observations demonstrated that NCT not only induced T cell reprogramming but also extensively impacted the composition and function of B cells in the TME.

## Crosstalk among tumor and immune cells

The TME consists of numerous cell types, and the importance of crosstalk between cancer and immune cells has been implicated in various biological processes associated with tumor development (*Chen*

*et al., 2021b*; *Lu et al., 2020*; *Raredon et al., 2019*). As depicted in *Figure 6—figure supplements 3a and b*, *Figure 6—figure supplement 4a*, the interactions between malignant cells and macrophages exhibited the strongest activity in both the control and NCT groups, highlighting the important role of macrophages in tumor immunology. Notably, we noted that the cell-to-cell communications among different cell types, especially between tumoral and immune cells such as cytotoxic CD8+ T, Treg, and memory B, were significantly strengthened in the NCT group. Specifically, we further investigated the ligand–receptor atlas within and between tumor cells and immune cells, which seemed to be quite reshaped by NCT (*Figure 6—figure supplements 3c and d*, *Figure 6—figure supplement 4b*). For example, MIF-CXCR4, whose activation usually promotes leukocyte recruitment (*Bernhagen et al., 2007*), was increasingly activated in the NCT group between malignant and memory B, CD4+ T, and cytotoxic CD8+ T, whereas it inhibited in macrophages. Meanwhile, MDK-NCL exhibited a similar activating phenotype with MIF-CXCR4, but its function in shaping the TME has never been reported. So, it might serve as a potential target of immune checkpoint inhibitor treatment in the future.

Given the abovementioned NCT-induced immune activation, which was characterized by CD8+ T with higher cytotoxicity and an increased proportion of class-switched memory B cells, these findings further clarified that NCT could ignite a strong intrinsic immune response toward tumor cells. However, the inhibitory interaction pairs LGALS9-CD44 and LGALS9-HAVCR2 were abnormally activated in the NCT group between malignant and several T cells or macrophages (*Anand et al., 2021*). Its exact role in such conditions still requires further exploration.

In summary, our study revealed that the LUAD tissues that have experienced NCT had a distinct landscape of intracellular interactions, which might provide new ideas for future research focusing on implementing immunotherapy in the comprehensive antitumor therapeutic regimen.

## Discussion

Although important advances in chemotherapy have reduced the mortality of cancer patients, the 5-year survival rate is still low, mainly due to the inherent or acquired mechanism of antitumor drug resistance (*Sharma, 2017*). Chemoresistance results from complex reprogramming processes, such as drug export/import, drug detoxification, DNA damage repair, and cell apoptosis. Recently, the correlation between metabolic regulation and chemoresistance has received great attention. More efforts are devoted to targeting cell metabolism to overcome chemoresistance (*Zhao et al., 2013*). The classic mechanism is to target the transport of anticancer drugs by increasing the activity of the efflux pump, such as the adenosine triphosphate (ATP) binding cassette (ABC) transporter. Cancer cells exhibit a special metabolic phenotype-aerobic glycolysis, quickly transporting and consuming glucose to produce ATP and promote drug efflux. PI3K/AKT pathway is activated by producing 3'-phosphorylated phosphoinositol, which is an important signaling pathway for lung cancer MDR (*Harding et al., 2019*). Glycolysis is beneficial to cancer cells by producing ATP faster, providing many intermediates for violent biosynthesis, maintaining redox balance, and creating a microenvironment with low immunity (*Tuerhong et al., 2021*). The combination therapy of shikonin + 2-DG could inhibit glycolytic phenotype, migration, and invasion by regulating the Akt/HIF1α/HK-2 signal axis (*Gupta et al., 2015*).

Normal and healthy cells mainly produce energy through OXPHOS. However, due to rapid cell growth and frequent division, cancer cells face impressive metabolic challenges, which force them to adjust their energy metabolism to meet these needs (*Vander Linden et al., 2021*). It is generally believed that cancer cells mainly obtain energy through glycolysis, which is named the Warburg effect. After chemotherapy, cancer cells change their metabolism from glycolysis to OXPHOS. This process is regulated by the SIRT1-PGC1α signaling pathway, thus increasing the resistance of cells to chemotherapy (*Vellinga et al., 2015*). Drug-resistant cancer cells can often be resensitized to anticancer treatments by targeting the metabolic pathways of import, catabolism, and synthesis of basic cell components (*Butler et al., 2013*). Recent studies have determined the cancer-promoting function of mitochondrial OXPHOS by regulating cell growth and redox homeostasis (*Zong et al., 2016*). Our study also found that after chemotherapy the glycolysis and OXPHOS of tumor cells were enhanced. This metabolic reprogramming may enable cancer cells to have higher proliferation, invasion, and metastasis capabilities.

Tumor ECs have high glycolytic metabolism, shunting intermediates to nucleotide synthesis. Blocking of the glycolysis activator PFKFB3 in EC cells does not affect tumor growth. Still, it reduces cancer cell invasion, intravascular, and metastasis by normalizing tumor blood vessels, thereby

improving blood vessel maturation and perfusion. PFKFB3 inhibition tightens the vascular barrier by reducing VE-cadherin endocytosis in ECs and reduces glycolysis to make cells more quiescent and adherent (by upregulating N-cadherin); it also reduces NF-κB signaling to reduce the expression of cancer cell adhesion molecules in ECs. PFKFB3 blockade therapy also improves chemotherapy for primary and metastatic tumors (*Cantelmo et al., 2016*).

Due to rapid cell growth and frequent division in tumor cells, cancer cells face impressive metabolic challenges, which force them to adjust energy metabolism to meet these needs, namely metabolic reprogramming (*Vander Heiden et al., 2009*). However, studies have shown that metabolic plasticity in tumors is contributed by the glycolytic phenotype (as explained by Warburg) and that mitochondrial energy reprogramming has recently been identified as a feature of tumors (*Jia et al., 2018*). Chemotherapy can increase SIRT1/PGC1α-dependent OXPHOS in tumor cells, thereby promoting the survival of colorectal tumors during treatment. This phenomenon was also observed in chemotherapy-exposed liver metastases, which strongly suggests that chemotherapy causes long-term changes in tumor metabolism that may interfere with drug efficacy (*Vellinga et al., 2015*). In addition, elevated glycolysis and OXPHOS promote epithelial-mesenchymal transition and cancer stem cell (CSC) phenotype in tumor cells (*Sciacovelli et al., 2016*). Therefore, recent research emphasizes the mixed glycolysis/OXPHOS phenotype rather than the phenotype that relies excessively on glycolysis to meet cellular energy requirements, thereby significantly promoting aggressiveness and treatment resistance (*Jia et al., 2018*). Chemotherapy has a significant effect on the metabolic reprogramming of tumor cells and profoundly affects stromal and immune cells' metabolism in the TME.

Theoretically, chemical drugs can inhibit tumorigenesis by blocking the proliferation of tumor cells or depositing in tumor cell apoptosis, but this unintentionally causes 'tissue damage'. The body will mistake this tumor-specific damage for normal tissue damage and then inevitably activate the tissue damage repair mechanism dominated by TAM (*Pathria et al., 2019*). The result of this effect is that tumors will grow rapidly, and patients will develop resistance to antitumor chemotherapy.

Meanwhile, as suggested by Parra et al., NCT exerted PD-L1 upregulation in non-small-cell lung cancer (NSCLC) patients. It increased the density of CD68+ macrophages, which were associated with better outcomes in both univariate and multivariate analyses (*Parra et al., 2018*). However, opposite results were also reported by Talebian et al. that NSCLC patients treated with radiotherapy, rather than a platinum-based standard-of-care chemotherapy, displayed a decrease in lymphoid cells and a relative increase in macrophages (*Talebian Yazdi et al., 2016*). Therefore, the role of TAMs in LUAD cells' response to chemotherapy still requires further investigation.

Our research reveals the remodeling effect of chemotherapy on TME. However, this study still has many limitations. Firstly, only nine samples were included in our study, and the number of samples is a defect of our study. Secondly, our study did not detect other causes of tumor heterogeneity, such as EGFR-mutant or ALK-translocated. We explored the possible impact of these factors and found that there was no significant difference in the expression of EGFR and other genes between the NCT group and the control group (*Supplementary file 1*). Thirdly, our data can only reflect the change in gene expression of various types of cells in the TME after chemotherapy. We cannot draw a direct conclusion on whether chemotherapy will benefit patients or not. These need further study in the future.

## Materials and methods
### Ethic statement
The Ethics Committee of Zhongshan Hospital, Fudan University, granted approval for our human research (approval no. B2021-230R), ensuring adherence to the Helsinki Declaration. Nine patients participated in this study after being fully informed of its purpose, procedures, risks, and benefits through detailed consent forms. Upon comprehension and agreement, they voluntarily signed the forms, consenting to tissue sample collection and the publication of our findings.

The experimental procedures involving animals were approved by the Animal Ethics Committee of Zhongshan Hospital, Fudan University (Shanghai, China), with approval no. 2021-718. All animals involved in this study were treated humanely and received standard care in accordance with the Guide for the Ethical Review of Animal Welfare (GB/T 35892-2018).

## Patients

The clinical samples of scRNA-seq came from patients diagnosed with LUAD, of which four cases received no treatment before surgery and five cases received chemotherapy (pemetrexed + cisplatin). These samples were donated by inpatients in the Department of Thoracic Surgery, Zhongshan Hospital of Fudan University (*Supplementary file 2*). After the LUAD tissue sample was taken, a small part was cut for paraffin sections, and the remaining tissue was dissociated into a single-cell suspension. 1 × $10^6$ cells were drawn from the single-cell suspension for single-cell RNA sequencing.

## Cell lines

The A549 cell line was acquired from the Cell Bank of the Chinese Academy of Sciences. Cell line identity was authenticated by short tandem repeat (STR) analysis and cultured in DMEM high-glucose medium (Hyclone, Logan, UT), supplemented with 10% fetal bovine serum (Hyclone), 100 U/mL penicillin, and 100 U/mL streptomycin. All cultures were incubated at 37°C in a humidified atmosphere containing 5% $CO_2$ and 95% air. Cells were used for no longer than 12 months before being replaced and were routinely tested for *Mycoplasma* to ensure the accuracy of experimental data. No cell line from the list of commonly misidentified cell lines was used.

## Preparation of single-cell suspensions

For each patient, as described above, we dissociated the LUAD tumor sample into a single-cell suspension and then took 1 × $10^6$ cells for single-cell RNA sequencing. We used the Tumor Dissociation Kit (Miltenyi Biotec, Gladbach, Germany) to digest tumor tissues with enzymes according to the manufacturer's instructions. In short, we first cut the LUAD tissue sample into small tissue pieces about 1 $cm^3$ with a surgical scalpel. We then transferred these small tissue pieces to the MACS C tube containing 4.7 mL DMEM serum-free medium, 200 µL Enzyme H, 100 µL Enzyme R, and 25 µL Enzyme A. After the tissue was incubated and digested in a constant temperature incubator 37°C for 1 hr, the tissue was mechanically separated using the MACS instrument. This procedure was repeated twice. After the tissue sample was dissociated, the sample was filtered with a 40 µm filter to remove the remaining large particles from the single-cell suspension. Centrifuge the suspension at 300 × *g* for 7 min, then discard the supernatant.

Next, we used red blood cell lysate (10×) (Sigma-Aldrich, St. Louis, MO) to remove red blood cells from the single-cell suspension. In short, add 1× Lysis Buffer to the centrifuge tube containing the single-cell pellet described above. The cell suspension was then incubated at room temperature for 15 min. To improve the quality of our samples, we also used a Dead Cell Removal Kit (Miltenyi Biotec) to ensure that the cell survival rate of our sequencing samples was >90%.

## The 10× scRNA-seq data analysis

The R version used in our scRNA-seq data analysis study is 3.6.1. The cell quality control criteria are as follows: (1) the number of expressed genes is less than 300 or greater than 5000; and (2) 10% or more of UMI is localized to mitochondrial or ribosomal genes. If they meet one of the criteria, the cells are excluded. After quality standardization, we applied the Seurat R package (*Macosko et al., 2015*) to analyze the scRNA-seq data. First, we converted the scRNA-seq data into Seurat identifiable objects, and then we used the 'FindVariableFeatures' function to find the first 2000 highly variable genes. After that, we applied principal component analysis to reduce the dimensionality of scRNA-seq data. The 'RunTSNE' function is used to perform t-distributed random neighborhood embedding (TSNE) to visualize various types of cells. The 'FindClusters' and 'FindAllMarkers' functions are used for cluster analysis of cell subclusters and detection of marker genes of cell subclusters.

Finally, according to the SingleR package (*Aran et al., 2019*), the CellMarker (http://bio-bigdata.hrbmu.edu.cn/CellMarker/) data set, and a previous report (*Lambrechts et al., 2018*), we annotated different cell types. Simultaneously, some new potential marker genes were verified through experiments.

## Analysis of subclusters of cells in LUAD

After preliminary classification and annotation of all cells, epithelial cells, stromal cells, and immune cells are extracted through the 'SubsetData' function. Then, we apply the 'FindClusters' and 'FindAllMarkers' functions to find the marker genes of each cell and perform dimensionality reduction clustering

on each extracted cell through TSNE. The subclusters are annotated by dominantly expressed cell markers published by previous researchers. To select the marker genes that meet the requirements, we set the following cutoff thresholds to reveal the marker genes of each cluster: adjusted p-value <0.01 and multiple Log2FC >0.5.

## Estimation of the CNVs

To estimate the initial CNV of each region, the R package 'scCancer' (*Guo et al., 2021*) was applied. The expression level of each cell was used as the original input file for calculating CNV. Immune cells served as a background reference for calculating the CNVs scores of other cells. In addition, the R package 'inferCNV' was used to quantify CNV in tumor cells as described previously (*Müller et al., 2016*).

## Definition of cell scores and signature

To evaluate the M1/M2 polarization state and pro-/anti-inflammatory potential of macrophages, we performed a GSVA. We retrieved gene sets related to the above functions from previous studies (*Sun et al., 2021*) and used them as references in this analysis.

We used the average expression of a published list of characteristic genes for T cell toxicity and exhaustion to define T cells' cytotoxicity, exhaustion, and costimulation scores.

## Identification of gene markers of malignant cells

We used the identified malignant cell marker genes in tumor cells to identify gene expression characteristics in malignant cells. Then, we performed unsupervised Non-negative Matrix Factorization (NMF) to reveal the malignant characteristics of tumor cells through the NMF R package (*Gaujoux and Seoighe, 2010*).

## Trajectory analysis

We used the monocle2 R package to analyze the trajectory of all cells to explore the trajectory progression of various types of cells in a single cell (*Qiu et al., 2017*). First, apply the function 'newCellDataSet' to construct a data object that the monocle 2R package can recognize. Afterward, the differentially expressed genes identified by the Seurat R package were selected for cell trajectory analysis. The 'reduceDimension' function was used to reduce the dimensionality. We used the 'orderCells' function to project cells on a pseudotime trajectory to show the trend of cell evolution. A state consisting of cells mainly derived from nonmalignant tissues in a cluster identified as epithelial cells was defined as 'root cells'.

## Analyses of metabolic pathways

To evaluate the activity of various metabolic pathways of each cell type, we applied the algorithm developed by *Xiao et al., 2019*. In short, the analysis of metabolic programs is based on the average expression level of metabolic genes across cell types to indirectly reflect the metabolic activity of cells.

A variety of environmental factors may potentially affect the metabolic reprogramming of tumors, such as chemotherapy, nutrient supply, and the environment where the cells are located. Therefore, exploring these factors and the cross-conversion between glycolysis and mitochondrial activity in various cells in the TME is essential for understanding the metabolic reprogramming of tumors.

We calculated the average gene expression levels in glycolysis and OXPHOS as indicators of glucose supply and mitochondrial activity, respectively. The data of genes that were responsive to the two groups of genes (known to be responsive to glycolysis and OXPHOS) used in the calculations were retrieved from the MsigDB database. At the same time, the cells were sorted by flow cytometry, and the contents of various metabolites were tested, in turn, to verify whether they were consistent with gene expression levels.

## Cell interaction network analysis

To study the cell-to-cell interactions between tumors and nonmalignant cells, immune cells, and stromal cells, we applied the R package 'CellChat' (*Jin et al., 2021*) and 'CellPhoneDB' Python package for analysis (*Patel et al., 2014*). The crosstalk analysis between cells through the 'CellChat' package was as follows: First, use the 'createCellChat' function to create a data set object that can be identified by

'CellChat'. Then, use 'aggregateNet', 'computeCommunProbPathway', and 'computeCommunProb' functions to automatically infer the possible cellular communication network between cells.Finally, the 'netVisual_aggregate', 'netVisual_bubble', and 'netVisual_signalingRole' functions were used to visualize the interaction between these cells. Then use the built-in parameters to apply the 'CellPhoneDB' R package.

## Immunohistochemistry and immunofluorescence

The paraffin-embedded lung cancer tissue sections were deparaffinized with xylene and rehydrated. Discard the blocking solution, add the primary antibody, and incubate overnight at 4°. After removing the primary antibody and washing thoroughly, add the secondary antibody to incubate for 1 hr, and then add DAB chromogenic reagent (Gene Tech, China) for color development. Finally, hematoxylin is used for nuclear dyeing.

As mentioned in the above immunohistochemistry experiment, the steps before incubating the primary antibody are the same. Incubate with the corresponding primary and secondary antibodies with green and red fluorescent dyes, respectively, and then use DAPI to stain the nuclei.

## Flow cytometry assay

Cells and APC-conjugated mouse anti-human CD45, FITC-conjugated mouse anti-human CD11b, BV421-conjugated mouse anti-human ARG1, as well as pe-cy-conjugated mouse anti-human CD86 (5 µL/$10^6$ cells; BD Biosciences) were incubated on ice for 30 min. Then, FACSAria III (BD Biosciences) was used to quantify the required cells, and FlowJo software (TreeStar, Woodburn, OR) was used to analyze the results.

## Animal experiments

In this experiment, we housed male athymic nude mice (BALB/cASlac-nu) in a specific pathogen-free environment. We mixed treated A549 cells and TAMs to make a 1:1 cell mixture at a cell concentration of $5 \times 10^6$ cells/mL. Take 0.05 mL of the mixed suspension of cells and Matrigel, and implant them into the lung thoracic cavity of nude mice for in situ tumor formation experiments.

Animals were sacrificed when one of the following signs of disease was observed: tumor ulceration (>0.5 cm), inability to move or eat, or serious injury. Changes in tumor size were detected using an optical imaging system for in vivo small animals (IVIS Spectrum, PerkinElmer, USA).

## Statistical analysis

The statistical tools, methods, and thresholds of each analysis are clearly described in the 'Results' or detailed in the legend or 'Materials and methods'.

## Acknowledgements

We thank OE Biotechnology Co., Ltd. (Shanghai, China) for helping us perform single-cell sequencing analysis of lung adenocarcinoma surgical samples. We thank International Science Editing Co. for providing language editing services for our papers.This research was funded by the Sailing Program of Shanghai (no. 22YF107300), the National Natural Science Foundation of China (no. 82203645), the China Postdoctoral Science Foundation (no. 2023M730655), the Natural Science Foundation of Shanghai, China (no. 22ZR1411900), and the Special Foundation for Supporting Biomedical Technology of Shanghai, China (no. 22S11900300).

## Additional information

### Funding

| Funder | Grant reference number | Author |
|---|---|---|
| China Postdoctoral Science Foundation | No. 2023M730655 | Yiwei Huang |

| Funder | Grant reference number | Author |
|---|---|---|
| Sailing Program of Shanghai | 22YF107300 | Yiwei Huang |
| The National Natural Science Foundation of China | 82203645 | Yiwei Huang |
| The Natural Science Foundation of Shanghai | 22ZR1411900 | Cheng Zhan |
| Special Foundation for Supporting Biomedical Technology of Shanghai | 22S11900300 | Cheng Zhan |

The funders had no role in study design, data collection and interpretation, or the decision to submit the work for publication.

## Author contributions

Yiwei Huang, Data curation, Visualization, Methodology, Writing - original draft, Writing – review and editing; Gujie Wu, Data curation, Validation, Methodology, Writing – review and editing; Guoshu Bi, Data curation, Software, Methodology; Lin Cheng, Data curation, Formal analysis; Jiaqi Liang, Visualization; Ming Li, Huan Zhang, Formal analysis; Guangyao Shan, Investigation; Zhengyang Hu, Software, Methodology; Zhencong Chen, Data curation; Zongwu Lin, Methodology; Wei Jiang, Supervision; Qun Wang, Software, Funding acquisition; Junjie Xi, Supervision, Project administration, Writing – review and editing; Shanye Yin, Supervision, Validation; Cheng Zhan, Resources, Data curation, Funding acquisition, Validation, Project administration, Writing – review and editing

## Author ORCIDs

Gujie Wu  https://orcid.org/0009-0002-6705-0857
Zhencong Chen  https://orcid.org/0000-0003-3555-0988
Shanye Yin  https://orcid.org/0000-0001-9116-5238
Cheng Zhan  https://orcid.org/0000-0001-8745-9276

## Ethics

The Ethics Committee of Zhongshan Hospital, Fudan University granted approval for our human research (Approval No. B2021-230R), ensuring adherence to the Helsinki Declaration. Nine patients participated in this study after being fully informed of its purpose, procedures, risks, and benefits through detailed consent forms. Upon comprehension and agreement, they voluntarily signed the forms, consenting to tissue sample collection and the publication of our findings.

The experimental procedures involving animals were approved by the Animal Ethics Committee of Zhongshan Hospital, Fudan University (Shanghai, China), with Approval No. 2021-718. All animals involved in this study were treated humanely and received standard care in accordance with the Guide for the Ethical Review of Animal Welfare (GB/T 35892-2018).

Reviewer #1 (Public review): https://doi.org/10.7554/eLife.95988.3.sa1
Author response https://doi.org/10.7554/eLife.95988.3.sa2

# Additional files

## Supplementary files

Supplementary file 1. The table shows the comparative analysis of differences in genetic testing results for all samples in this study.

Supplementary file 2. Clinical characteristics of the patients included in this study.

MDAR checklist

## Data availability

The single-cell sequencing data used in this study and other experimental data can be obtained from the figshare https://doi.org/10.6084/m9.figshare.24797265.

The following dataset was generated:

| Author(s) | Year | Dataset title | Dataset URL | Database and Identifier |
|-----------|------|---------------|-------------|-------------------------|
| Yiwei H | 2024 | Unveiling chemotherapy-induced immune landscape remodeling and metabolic reprogramming in lung adenocarcinoma by scRNA-sequencing | https://figshare.com/articles/dataset/dx_doi_org_10_6084_m9_figshare_6025748/6025748 | figshare, 10.6084/m9.figshare.24797265 |

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
