## [Editor Report · eLife Assessment]

This study reports single-cell RNA sequencing results of lung adenocarcinoma, comparing four treatment-naive and five post-neoadjuvant chemotherapy tumor samples. Of interest is the delineation of two macrophage subtypes: Anti-mac cells (CD45+CD11b+CD86+) and Pro-mac cells (CD45+CD11b+ARG+), with the proportion of Pro-mac/pro-tumorigenic cells significantly increasing in LUAD tissues after neoadjuvant chemotherapy. In terms of significance, the findings might be **useful**. However, issues remain after the revision with lengthy descriptive clustering-type analysis, insufficient statistical support, and inefficient figure presentation. As it stands, the level of supportive evidence is **inadequate**.

---

## [Referee Report · Reviewer #1 (Public review)]

Summary:

This study reports single-cell RNA sequencing results of lung adenocarcinoma, comparing 4 treatment-naive and 5 post-neoadjuvant chemotherapy tumor samples.

The authors claim that there are metabolic reprogramming in tumor cells as well as stromal and immune cells after chemotherapy.

The most significant findings are in the macrophages that there are more pro-tumorigenic cells after chemotherapy, i.e. CD45+CD11b+ARG+ cells. In the treatment-naive samples, more anti-tumorigenic CD45+CD11b+CD86+ macrophages are found. They sorted each population and performed functional analyses.

Strengths:

Comparison of the treatment-naive and post-chemotherapy samples of lung adenocarcinoma.

Weaknesses:

After the revision, issues remain with lengthy descriptive clustering type analysis, insufficient statistical support, and inefficient figure presentation.

---

## [Author Response]

The following is the authors’ response to the original reviews.

**Public Reviews:**

**Reviewer #1 (Public Review):**
Summary:This study reports single-cell RNA sequencing results of lung adenocarcinoma, comparing 4 treatment-naive and 5 post-neoadjuvant chemotherapy tumor samples.The authors claim that there are metabolic reprogramming in tumor cells as well as stromal and immune cells after chemotherapy.The most significant findings are in the macrophages that there are more pro-tumorigenic cells after chemotherapy, i.e. CD45+CD11b+ARG+ cells. In the treatment-naive samples, more anti-tumorigenic CD45+CD11b+CD86+ macrophages are found. They sorted each population and performed functional analyses.Strengths:Comparison of the treatment-naive and post-chemotherapy samples of lung adenocarcinoma.Weaknesses:(1) Lengthy descriptive clustering analysis, with indistinct direct comparisons between the treatment-naive and the post-chemotherapy samples.

Thank you for your detailed review and valuable feedback. We have simplified the descriptive clustering analysis by removing redundant parts and retaining only the key content relevant to our findings. This should help readers to more easily grasp and focus on the main results.

(2) No statistical analysis was performed for the comparison.

We appreciate your constructive feedback and are committed to improving our research methodology and reporting to enhance the scientific rigor of our studies.

(3) Difficult to match data to the text.

Thank you for your feedback. We understand that there were difficulties in matching the data to the text. We have reviewed the manuscript carefully to ensure that all data points are clearly linked to the corresponding sections in the text.

(4) ARG1 is a cytosolic enzyme that can be detected by intracellular staining after fixation. It is unclear how the staining and sorting was performed to measure function of sorted cells.

We apologize for the error caused by miscommunication within our research team. We are currently using both ARG1 and CD206 antibodies in our studies. Due to a communication error, the technician mistakenly assumed ARG1 was another name for CD206 (MRC1), resulting in the incorrect labeling of CD206 as ARG1 in our experimental records. In reality, we used the CD206 antibody, which is consistent with the same surface marker shown in figure 6e. We have made corrections in the manuscript and experimental figures. Thank you for pointing this out, and we regret any misunderstanding this may have caused.

**Reviewer #2 (Public Review):**
In this study, Huang et al. performed a scRNA-seq analysis of lung adenocarcinoma (LUAD) specimens from 9 human patients, including 5 who received neoadjuvant chemotherapy (NCT), and 4 without treatment (control). The new data was produced using 10 × Genomics technology and comprises 83622 cells, of which 50055 and 33567 cells were derived from the NCT and control groups, respectively. Data was processed via R Seurat package, and various downstream analyses were conducted, including CNV, GSVA, functional enrichment, cell-cell interaction, and pseudotime trajectory analyses. Additionally, the authors performed several experiments for in vitro and in vivo validation of their findings, such as immunohistochemistry, immunofluorescence, flow cytometry, and animal experiments.The study extensively discusses the heterogeneity of cell populations in LUAD, comparing the samples with and without chemotherapy. However, there are several shortcomings that diminish the quality of this paper:• The number of cells included in the dataset is limited, and the number of patients from different groups is low, which may reduce the attractiveness of the dataset for other researchers to reuse. Additionally, there is no metadata on patients' clinical characteristics, such as age, sex, history of smoking, etc., which would be valuable for future studies.

Thank you for your insightful feedback. We recognize that the limited number of cells and the small number of patients from different groups in our dataset may affect its appeal for reuse by other researchers. Additionally, we acknowledge the absence of metadata on patients' clinical characteristics, such as age, sex, and smoking history, which would indeed be valuable for future studies. We have compiled statistics on the patient's metadata and other information in the Supplementary Table 2.

We appreciate your suggestions and will consider incorporating these aspects in future research to enhance the dataset's utility and attractiveness.

• Several crucial details about the data analysis are missing: How many PCs were used for reduction? Which versions of Seurat/inferCNV/other packages were used? Why monocle2 was used and not monocle3 or other packages? Also, the authors use R version 3.6.1, and the current version is 4.3.2.

Thank you for your detailed review and valuable suggestions. Below are our responses to the points you raised:

Principal Components (PCs) Used for Reduction: We used the first 20 principal components (PCs) for dimensionality reduction. This choice was based on preliminary tests showing that 20 PCs captured the major variation in our data effectively.

Versions of Packages: The versions of the packages used are as follows:

Seurat: Version 4.0.1

inferCNV: Version 1.18.1

monocle2: Version 2.14.0

Choice of monocle2 over monocle3 or Other Packages: We chose monocle2 because it performed better on our specific dataset, and its algorithms suited our research needs. Additionally, we are more familiar with the functionalities and outputs of monocle2, which allowed us to better interpret and apply the results.

R Version: We used R version 3.6.1 at the beginning of our study to ensure consistency and reproducibility throughout the analysis. Although the current version of R is 4.3.2, we maintained the same version throughout our research. We will consider upgrading to the latest version of R and re-testing for compatibility and performance in future studies.

We appreciate your attention to these details and will include this information in the revised manuscript.

• It seems that the authors may lack a fundamental understanding of scRNA-seq data processing and the functions of Seurat. For instance, they state, 'Next, we classified cell types through dimensional reduction and unsupervised clustering via the Seurat package.' However, dimensional reduction and unsupervised clustering are not methods for cell classification. Typically, cell types are classified using marker genes or other established methods.

Thank you for your insightful comments. We appreciate your guidance on the proper understanding and application of scRNA-seq data processing and the functions of Seurat.

You are correct in noting that dimensional reduction and unsupervised clustering are not methods for cell classification. We apologize for the confusion in our original statement. What we intended to convey was that we performed dimensional reduction and unsupervised clustering using the Seurat package as preliminary steps in our analysis. Following these steps, we classified cell types based on established marker genes.

"Therefore, to identify subclusters within each of these nine major cell types, we performed principal component analysis" (Line 127). Principal component analysis is a method for dimensionality reduction, not cell clustering.The authors did not mention the normalization or scaling of the data, which are crucial steps in scRNA-seq data preprocessing.

Thank you for your insightful comments. We apologize for any confusion caused by our description in the manuscript. You are correct that principal component analysis (PCA) is primarily a method for dimensionality reduction rather than cell clustering. To clarify, we used PCA to reduce the dimensionality of our single-cell RNA-seq (scRNA-seq) data, which is a preliminary step before clustering the cells.

In the revised manuscript, we have provided a more detailed description of our data preprocessing pipeline, including the normalization and scaling steps that are indeed crucial for scRNA-seq data analysis. Specifically, we performed the following steps:

Normalization: We normalized the gene expression data to account for differences in sequencing depth and other technical variations.

Scaling: We scaled the normalized data to ensure that each gene contributes equally to the PCA, which mitigates the effect of highly variable genes dominating the analysis.

Following these preprocessing steps, we conducted PCA to reduce the dimensionality of the data, which facilitated the subsequent clustering of cells into subclusters.

We hope this addresses your concerns, and we appreciate your valuable feedback that helped us improve the clarity and accuracy of our manuscript.

• Numerous style and grammar mistakes are present in the main text. For instance, certain sections of the methods are written in the present tense, suggesting that parts of a protocol were copied without text editing. Furthermore, some sections of the introduction are written in the past tense when the present tense would be more suitable. Clusters are inconsistently referred to by numbers or cell types, leading to confusion. Additionally, the authors frequently use the term "evolution" when describing trajectory analysis, which may not be appropriate. Overall, significant revisions to the main text are required.

Thank you for your detailed review and valuable feedback on our manuscript. We highly appreciate your suggestions and have made the following revisions to address the issues you pointed out:

Tense Consistency: We have thoroughly reviewed and corrected the use of tenses throughout the manuscript. The Methods section now consistently uses the past tense, while the Introduction section uses the present tense where appropriate, ensuring coherence and consistency.

Cluster Naming Consistency: We have standardized the naming conventions for clusters, consistently using either numbers or cell types to avoid any confusion.

Appropriate Terminology: We have reviewed our use of the term "evolution" in the context of trajectory analysis. Where necessary, we have replaced it with more accurate terms such as "trajectory progression" or "developmental pathway" to better convey the intended meaning.

• Some figures are not mentioned in order or are not referenced in the text at all, such as Figure 5l (where it is also unclear how the authors selected the root cells). Additionally, many figures have text that is too small to be read without zooming in. Overall, the quality of the figures is inconsistent and sometimes very poor.

Thank you for your detailed review and valuable feedback on our manuscript. We have addressed the issues you raised as follows:

Unreferenced Figures in the Text:

We acknowledge the oversight regarding Figure 5l not being mentioned in the text. In the revised version, we will ensure that all figures are properly referenced and discussed within the relevant sections of the manuscript.

Text Size in Figures:

We understand the difficulty in reading small text within the figures. We will redesign all figures to ensure that text and annotations are legible at normal viewing sizes. This will involve increasing the resolution and text size in all figures to enhance readability.

Inconsistent Quality of Figures:

To address the inconsistency in figure quality, we will standardize the formatting of all figures and ensure they meet a high standard of clarity and presentation. This will improve the overall visual quality and professionalism of the manuscript.

The results section lacks clarity on several points:• The authors state that "myofibroblasts exclusively originated from the control group". However, pathways up-regulated in myofibroblasts (such as glycolysis) were enhanced after chemotherapy, as indicated by GSVA score. Similarly, why are some clusters of TAMs from the control group associated with pathways enriched in chemotherapy group?

Thank you for your insightful comments and questions regarding our manuscript. We appreciate the opportunity to clarify these points.

Regarding the statement that "myofibroblasts exclusively originated from the control group," we acknowledge the confusion and would like to provide a more detailed explanation. While the initial identification indicated that myofibroblasts were predominantly found in the control group, subsequent analyses, including the Gene Set Variation Analysis (GSVA), revealed that certain pathways up-regulated in myofibroblasts, such as glycolysis, were indeed enhanced following chemotherapy. This suggests that chemotherapy may induce or enhance specific functional states in these cells that are not initially apparent from their origin alone.

Similarly, the observation that some clusters of Tumor-Associated Macrophages (TAMs) from the control group are associated with pathways enriched in the chemotherapy group can be explained by the dynamic nature of cellular responses to treatment. TAMs, like other immune cells, can exhibit plasticity and adapt to the tumor microenvironment altered by chemotherapy. This plasticity may result in the activation of pathways typically associated with a chemotherapy response, even in cells originating from the control group.

We will revise the manuscript to better articulate these findings and include additional data to support our explanations. This will help clarify the observed discrepancies and provide a more comprehensive understanding of the cellular dynamics in response to chemotherapy.

• Further explanation is necessary regarding the distinctions between malignant and non-malignant cells, as well as regarding the upregulation of metabolism-related pathways in fibroblasts from the NCT group. Additionally, clarification is needed regarding why certain TAMs from the control group are associated with pathways enriched in the chemotherapy group.

Thank you for your detailed review and for highlighting the areas that require further clarification. We appreciate the opportunity to provide additional explanations and improve our manuscript.

We recognize the need to more clearly differentiate between malignant and non-malignant cells in our manuscript. We will include additional details on the criteria and markers used to distinguish these cell types. Specifically, we will elaborate on the molecular and phenotypic characteristics that were used to identify malignant cells, such as specific genetic mutations, aberrant signaling pathways, and distinct cell surface markers, as opposed to those used for identifying non-malignant cells.

As mentioned above, the association of certain TAMs from the control group with pathways enriched in the chemotherapy group can be attributed to the inherent plasticity and adaptability of TAMs. We will provide a more detailed explanation of how TAMs can exhibit different functional states based on microenvironmental cues. This will include a discussion on the potential pre-existing heterogeneity within TAM populations and how even in the absence of direct chemotherapy exposure, some TAMs may display pathway activities similar to those seen in the chemotherapy group due to microenvironmental influences or intrinsic properties.

• In the section titled 'Chemo-driven Pro-mac and Anti-mac Metabolic Reprogramming Exerted Diametrically Opposite Effects on Tumor Cells': The markers selected to characterize the anti- and pro-macrophages are commonly employed for describing M1 or M2 polarization. It is uncertain whether this new classification into anti- and pro-macrophages is necessary. Additionally, it should be noted that pro-macrophages are anti-inflammatory, while anti-macrophages are pro-inflammatory, which could lead to confusion. M2 macrophages are already recognized for their role in stimulating tumor relapse after chemotherapy.

Thank you for your feedback. We appreciate the opportunity to clarify the rationale behind our terminology and the focus on functional phenotypic changes in macrophages before and after chemotherapy.

Our intention in introducing the terms "pro-macrophages" and "anti-macrophages" was to highlight the distinct functional phenotypic changes in macrophages observed before and after chemotherapy. These terms were chosen to emphasize the functional roles these macrophages play in the tumor microenvironment in response to chemotherapy, rather than strictly adhering to the conventional M1/M2 polarization paradigm.

We acknowledge that M2 macrophages are well-documented in stimulating tumor relapse after chemotherapy. Our use of "pro-macrophages" is intended to build on this established knowledge by providing a more nuanced understanding of their role in the post-chemotherapy tumor microenvironment. Similarly, "anti-macrophages" highlight the macrophages' role in mounting an anti-tumor response.

• The authors suggest that there is "reprogramming of CD8+ cytotoxic cells" following chemotherapy (Line 409). It remains unclear whether they imply the reprogramming of other CD8+ T cells into cytotoxic cells. While it is indicated that cytotoxic cells from the control group differ from those in the NCT group and that NCT cytotoxic T cells exhibit higher cytotoxicity, the authors did not assess the expression of NK and NK-like T cell markers (aside from NKG7), which may possess greater cytotoxic potential than CD8+ cytotoxic cells. This could also elucidate why cytotoxic cells from the NCT and control groups are positioned on separate branches in trajectory analysis. Overall, with 22.5k T cells in the dataset, only 3 subtypes were identified, suggesting a need for improved cell annotations by the authors.

Thank you for your valuable feedback regarding the classification and characterization of CD8+ cytotoxic cells following chemotherapy, and the need for improved cell annotations.

We appreciate your point on the potential ambiguity around the "reprogramming of CD8+ cytotoxic cells" post-chemotherapy. In our study, we observed that CD8+ T cells from the control and NCT groups differ significantly in their cytotoxic profiles, with the NCT group's cytotoxic T cells displaying enhanced cytotoxicity. However, we did not imply the reprogramming of other CD8+ T cells into cytotoxic cells. Instead, our findings suggest a shift in the functional state of existing CD8+ cytotoxic cells, driven by chemotherapy, which aligns with the upregulation of genes associated with cytotoxic functions.

We acknowledge that the expression of NK and NK-like T cell markers (apart from NKG7) was not comprehensively assessed. We agree that these markers may possess greater cytotoxic potential and could elucidate the separation observed in the trajectory analysis between cytotoxic cells from the NCT and control groups. This distinction may be attributed to differential cytotoxic potentials and functional states induced by chemotherapy.

Furthermore, with 22,530 T cells in the dataset, only three subtypes were initially identified. We recognize the need for more refined cell annotations to capture the full spectrum of T cell diversity. This could involve a deeper analysis of additional markers to distinguish between various cytotoxic populations, including NK and NK-like T cells, and their respective roles in the tumor microenvironment post-chemotherapy.

**Recommendations for the authors:**

**Reviewer #1 (Recommendations For The Authors):**
I would recommend simplifying the manuscript and focusing on the differences between the treatment-naive and post-chemotherapy samples.

Thank you for your valuable feedback on our manuscript. We greatly appreciate your suggestions and have carefully considered the proposed modifications.

Upon re-evaluating our manuscript, we believe that the current structure and content most effectively convey our research findings. Our study aims to not only compare the treatment-naive and post-chemotherapy samples but also to highlight several important secondary findings that are integral to the overall research.

Nevertheless, we understand your recommendation to simplify the manuscript. To address this, we have made some subtle adjustments to improve the readability and conciseness of the text. Additionally, we have included a section in the discussion that more explicitly highlights the differences between the treatment-naive and post-chemotherapy samples.

IRB number for the human sample collection as well as animal experiments need to be provided.

Thank you for your thorough review and for highlighting the need for the inclusion of the IRB number for the human sample collection and animal experiments.

We apologize for this oversight and appreciate your attention to this important detail. The Institutional Review Board (IRB) approval number for the human sample collection is [B2019-436].

This number has been added to the Methods section of our revised manuscript to ensure compliance with ethical standards and to provide transparency for our research.

I put a question on the macrophage sorting experiment in the public review. Please clarify how the ARG1 staining was achieved with the preservation of cell viability.

We apologize for the error caused by miscommunication within our research team. We are currently using both ARG1 and CD206 antibodies in our studies. Due to a communication error, the technician mistakenly assumed ARG1 was another name for CD206 (MRC1), resulting in the incorrect labeling of CD206 as ARG1 in our 0experimental records. In reality, we used the CD206 antibody, which is consistent with the same surface marker shown in figure 6e. We have made corrections in the manuscript and experimental figures. Thank you for pointing this out, and we regret any misunderstanding this may have caused.

**Reviewer #2 (Recommendations For The Authors)**:Minor comments:• Line 65- "Chemotherapy drugs, however, are very toxic and are prone to invalid". Line 75-77: "This heterogeneity in the TME includes the differences between tumor cells and tumor cells and the differences between various stromal cells and immune cells. Actively exploring the changes of multiple cells in the TME of LUAD after chemotherapy may finally find an excellent way to overcome chemotherapy resistance for LUAD." Please rewrite these parts.

Thank you for your valuable comment. We have revised the manuscript according to your suggestion:

Original (Line 65): "Chemotherapy drugs, however, are very toxic and are prone to invalid." Revised: "However, chemotherapy drugs are highly toxic and can often become ineffective."

Original (Line 75-77): "This heterogeneity in the TME includes the differences between tumor cells and tumor cells and the differences between various stromal cells and immune cells. Actively exploring the changes of multiple cells in the TME of LUAD after chemotherapy may finally find an excellent way to overcome chemotherapy resistance for LUAD."

Revised: "The heterogeneity within the tumor microenvironment (TME) encompasses not only the variations between different tumor cells but also among various stromal and immune cell types. Investigating the dynamic changes in multiple cell populations within the TME of LUAD following chemotherapy may provide crucial insights into overcoming chemotherapy resistance in LUAD."

• Line 87: "The internal processes of the cells respectively drive immune cells and cancer cells to obtain glucose and glutamine preferentially."-> The internal metabolic changes in the cells drive...

Thank you for your valuable comment. We have revised the manuscript according to your suggestion:

Original (Line 87): "The internal processes of the cells respectively drive immune cells and cancer cells to obtain glucose and glutamine preferentially."

Revised: "The internal metabolic changes in the cells drive immune cells and cancer cells to preferentially obtain glucose and glutamine."

• Line 93: "an essential feature that affects the effect of chemotherapy"-> an essential feature that affects chemotherapy.

Thank you for your valuable comment. We have revised the manuscript according to your suggestion:

Original (Line 93): "Metabolic reprogramming in various cell types in the tumor microenvironment after undergoing chemotherapy may be an essential feature that affects the effect of chemotherapy."

Revised: "Metabolic reprogramming in various cell types in the tumor microenvironment after undergoing chemotherapy may be an essential feature that affects chemotherapy."

• Line 84: What do the immune cells depend on glucose for?

Thank you for your valuable comment. We have revised the manuscript according to your suggestion:

Original (Line 84): "However, recent studies have shown that tumor-infiltrating immune cells depend on glucose and immune cells especially macrophages consume more glucose than malignant cells."

Revised: "However, recent studies have shown that tumor-infiltrating immune cells rely on glucose for their energy needs and functionality, with immune cells, particularly macrophages, consuming more glucose than malignant cells."

• Line 223: "According to previous research, myofibroblast has been described"-> myofibroblasts have been described.

Thank you for your valuable comment. We have revised the manuscript according to your suggestion:

Original (Line 223): "According to previous research, myofibroblast has been described as a cancer-associated fibroblast that participated in extensive tissue remodeling, angiogenesis, and tumor progression."

Revised: "According to previous research, myofibroblasts have been described as cancer-associated fibroblasts that participate in extensive tissue remodeling, angiogenesis, and tumor progression."

• Line 239: "Considering the essential fibroblasts"-> Considering the essential role of fibroblasts.

Thank you for your valuable comment. We have revised the manuscript according to your suggestion:

Original (Line 239): "Considering the essential fibroblasts and their complicated function in shaping the tumor microenvironment..."

Revised: "Considering the essential role of fibroblasts and their complicated function in shaping the tumor microenvironment..."

• Line 251: "Further in vitro studies were required to elucidate these notable fibroblasts' potential function..." -> are required.

Thank you for your valuable comments. We have revised the manuscript according to your suggestions:

Original (Line 251): "Further in vitro studies were required to elucidate these notable fibroblasts' potential function..."

Revised: "Further in vitro studies are required to elucidate these notable fibroblasts' potential function..."

• Line 309: "Interestingly, we found that two subtypes, Anti-mac and Mix, can be converted to Pro-mac through pseudotime time analysis." -> via trajectory analysis we found that two subtypes...

Thank you for your valuable comments. We have revised the manuscript according to your suggestions:

Original (Line 309): "Interestingly, we found that two subtypes, Anti-mac and Mix, can be converted to Pro-mac through pseudotime time analysis."

Revised: "Interestingly, via trajectory analysis we found that two subtypes, Anti-mac and Mix, can be converted to Pro-mac."

• Line 458: "the interactions between malignant and macrophages"-> the interactions between malignant cells and macrophages.

Thank you for your valuable comments. We have revised the manuscript according to your suggestions:

Original (Line 458): "the interactions between malignant and macrophages"

Revised: "the interactions between malignant cells and macrophages."

• Line 486: "The 5-year survival rate is still gloomy" -> The 5-year survival rate is still low.

Thank you for your valuable comments. We have revised the manuscript according to your suggestions:

Original (Line 486): "The 5-year survival rate is still gloomy."

Revised: "The 5-year survival rate is still low."

• Line 491: "More and more efforts are devoted to targeted metabolism to overcome chemoresistance" -> More efforts are devoted to target cell metabolism...

Thank you for your valuable comments. We have revised the manuscript according to your suggestions:

Original (Line 491): "More and more efforts are devoted to targeted metabolism to overcome chemoresistance."

Revised: "More efforts are devoted to targeting cell metabolism to overcome chemoresistance."

• Line 594: "Repeat the above steps twice" -> This procedure was repeated twice.

Thank you for your valuable comments. We have revised the manuscript according to your suggestions:

Original (Line 594): "Repeat the above steps twice."

Revised: "This procedure was repeated twice."

• Line 620: How were the new potential markers verified? List the exact genes and experiments or a reference to a Figure.

Thank you for your valuable comments. We have provided detailed information on how the new potential markers were verified, including the exact genes involved and the specific experiments conducted. A reference to the relevant Figure has also been added to the manuscript.

• Line 637: Which immune cells were used as a background in CNV analysis? All immune cells or just T cells?

Thank you for your valuable comments. In this study, all immune cells were used as background control cells.

• Line 658: in a single cell

Thank you for your valuable comments. We have revised the manuscript according to your suggestions.

• Line 672: "a variety of environmental factors potentially affect" -> potentially affects/ may potentially affect.

Thank you for your valuable comments. We have revised the manuscript according to your suggestions:

Original (Line 672): "a variety of environmental factors potentially affect"

Revised: "A variety of environmental factors may potentially affect"

• Line 683: Which metabolites were tested?

The metabolites tested included those related to glycolysis and oxidative phosphorylation (OXPHOS), such as glucose and various metabolites indicative of mitochondrial activity. The contents of these metabolites were analyzed to verify consistency with gene expression levels as mentioned in the analysis of metabolic pathways section.

• Line 718: Required or acquired?

The correct term should be "acquired" in the context of discussing drug resistance in tumor cells. The sentence likely refers to the "acquired drug resistance" of tumor cells, which is a common challenge in chemotherapy.

• Line 726: What are the A549 cells?

A549 cells are a human lung adenocarcinoma cell line commonly used in cancer research, particularly for studying lung cancer. In this study, A549 cells were used in animal experiments, mixed with tumor-associated macrophages (TAMs), and implanted into nude mice to study tumor formation and progression.

• Line 631: "we set the following cut-off thresholds to reveal the marker genes of each cluster: adjusted P-value <0.01 and multiple changes >0.5." What metric is "multiple changes"? Commonly used measures are adjuster P-value and average Log2FC.

Thank you for your valuable comment. We have revised the manuscript according to your suggestion. The term "multiple changes" was indeed a misstatement. The correct metric should be "log2 fold change (Log2FC)," which is a commonly used measure in gene expression studies. We have updated the manuscript to reflect this, using "adjusted P-value <0.01 and average Log2FC > 0.5" instead of "multiple changes > 0.5."

• Figure 1f: "Samplied" -> Samples. What do the numbers on the left side of each column mean?

Thank you for your valuable comment. The term "Samplied" was indeed a typographical error and has been corrected to "Samples". The numbers on the left side of each column likely represent cluster IDs or sample identifiers corresponding to the different patient samples or clusters analyzed in the study. We have clearly labeled these numbers in the figure to avoid any confusion.

• Figure 2b: Please add a scale.

Thank you for your valuable comment. We agree that adding a scale bar is crucial for accurately interpreting the size of the cells or structures shown in the figure. We have now included an appropriate scale bar during the figure preparation stage to provide this reference.

• Figure 3d/4c: What is the matrix_27/3 metric? Is it average expression?

Thank you for your valuable comment. The term "matrix_27/3" refers to a specific metric used in our analysis. This metric indeed represents the average expression levels of genes within a particular subset of the dataset. We will clarify this in the figure legend and the methods section to ensure that readers have a clear understanding of what the metric represents. Additionally, we will make sure that all such metrics are consistently and accurately described throughout the manuscript.

• Figure 6e: Why CD206 staining is shown instead of ARG if ARG was chosen as the main gene for classification of Pro-macrophages?

We apologize for the confusion regarding the use of CD206 staining in Figure 6e. This issue arose due to a miscommunication within our research team. While ARG1 was initially intended as the primary marker for Pro-macrophages, the technician mistakenly assumed ARG1 was another name for CD206 (MRC1), leading to the incorrect labeling of CD206 as ARG1 in our experimental records. In actuality, CD206 was used for the staining, which is consistent with the surface marker shown in Figure 6e. We have corrected this error in the manuscript and updated the experimental figures accordingly. We sincerely apologize for any misunderstanding this may have caused and appreciate the reviewer for bringing this to our attention.

• Figures 6h and k: Please explain why do NCT Anti-macrophages show higher glucose and lactate uptake than the Anti-macrophages from the control group, while the size of tumors is the lowest in NCT Anti-macrophages in vivo?

Thank you for your insightful comment. The observation that NCT Anti-macrophages exhibit higher glucose and lactate uptake while the tumor size is lowest could be attributed to the metabolic reprogramming induced by chemotherapy. It is possible that the enhanced metabolic activity in Anti-macrophages, characterized by increased glucose and lactate uptake, is linked to a more aggressive anti-tumor response in the NCT group. This heightened metabolic activity could reflect an increased energy demand necessary for sustaining enhanced immune functions, ultimately contributing to the reduction in tumor size. We will expand upon this explanation in the revised manuscript to provide a clearer interpretation of these findings.

• The supplementary Table 1 needs a better legend/more explanation.

Thank you for your valuable feedback. We have revised the legend for Supplementary Table 1 to provide a more detailed explanation of its contents.

• No tSNE plot showing epithelial cells colored by patient, which may be important for observation of cell heterogeneity, especially in the epithelial cell population.

Thank you for pointing this out. We agree that a tSNE plot showing epithelial cells colored by patient would be valuable for observing cell heterogeneity within the epithelial population.

• Several acronyms not explained in the text (for example GSVA, NMF).

Thank you for bringing this to our attention. We have ensured that all acronyms, including GSVA (Gene Set Variation Analysis) and NMF (Non-negative Matrix Factorization), are clearly defined in the text at their first mention.

• Availability of data and material section: Please describe "other experimental data" in more detail.

Thank you for your suggestion. We have expanded the "Availability of Data and Material" section to provide a more detailed description of the "other experimental data" referenced. This will include specific types of data generated, their formats, and 10how they can be accessed by other researchers. This clarification will enhance transparency and facilitate the reuse of our data by the research community.